



# Comparison of eddy covariance $CO_2$ and $CH_4$ fluxes from mined and recently rewetted sections in a NW German cutover bog

David Holl[1], Eva-Maria Pfeiffer[1], and Lars Kutzbach[1]

[1]Institute of Soil Science, Center for Earth System Research and Sustainability (CEN), Universität Hamburg, Hamburg, Germany

**Correspondence:** David Holl (david.holl@uni-hamburg.de)

**Abstract.** With respect to their role in the global carbon cycle, natural peatlands are characterized by their ability to sequester atmospheric carbon. This trait is strongly connected to the water regime of these ecosystems. Large parts of the soil profile in natural peatlands are water-saturated, leading to anoxic conditions and to a diminished decomposition of plant litter. In functioning peatlands, the rate of carbon fixation by plant photosynthesis is larger than the decomposition rate of dead organic

material. Over time, the amount of carbon that remains in the soil and is not converted back to carbon dioxide grows. Land use of peatlands often goes along with water level manipulations and thereby with alterations of carbon flux dynamics. In this study, carbon dioxide ($CO_2$) and methane ($CH_4$) flux measurements from a bog site in NW Germany that has been heavily degraded by peat mining are presented. Two contrasting types of management have been implemented at the site: (1) drainage during ongoing peat-harvesting on one half of the central bog area and (2) rewetting on the other half that had been taken out of use

shortly before measurements commenced. The presented two-year data set was collected with an eddy covariance (EC) system set up on a central railroad dam that divides the two halves of the (former) peat harvesting area. We used footprint analysis to split the obtained $CO_2$ and $CH_4$ flux time series into data characterizing the gas exchange dynamics of both contrasting land use types individually. The time series gaps resulting from data division were filled using the response of artificial neural networks (ANNs) to environmental variables, footprint variability and fuzzy transformations of seasonal and diurnal cyclicity.

We used the gap-filled gas flux time series from two consecutive years to evaluate the impact of rewetting on the annual vertical carbon balances of the cutover bog. Rewetting had a considerable effect on the annual carbon fluxes and led to increased $CH_4$ and decreased $CO_2$ release.

The larger relative difference between cumulative $CO_2$ fluxes from the rewetted ($22 \pm 7$ mol m$^{-2}$ a$^{-1}$) and drained ($13 \pm 6$ mol m$^{-2}$ a$^{-1}$) section occurred in the second observed year when rewetting apparently reduced $CO_2$ emissions by 40 %. The

absolute difference in annual $CH_4$ flux sums was more similar between both years while the relative difference of $CH_4$ release between the rewetted ($0.83 \pm 0.15$ mol m$^{-2}$ a$^{-1}$) and drained ($0.45 \pm 0.11$ mol m$^{-2}$ a$^{-1}$) section was larger in the first observed year indicating a maximum increase of annual $CH_4$ release of 84 % caused by rewetting; at this particular site during the study period.





# 1 Introduction

Peatlands are wetland ecosystems that accumulate peat under water-saturated soil conditions. Peat formation is the result of an imbalance between production and decomposition of organic matter. For a peatland to qualify as a mire, the accumulation of peat has to be ongoing. The term peatland is defined broader and refers to soils that include an at least 30 cm thick peat horizon

– with or without ongoing peat accumulation. Concerning long-term carbon sequestration, no other terrestrial ecosystems are as efficient as mires. Although peatlands cover only 3 % (400 million ha) of the Earth's land surface, they store 550 Gt carbon (Yu et al., 2010), which equals the amount of carbon (C) stored in the entire terrestrial biomass and represents twice as much C as sequestered in the Earth's forests respectively. Peatlands are characterized by complex interactions between vegetation, hydrology and peat and are therefore vulnerable to alterations of these factors by men or climate change. Traditional

land use practices in peatlands are commonly paralleled by interference with the ecosystems' water regimes. Hydrological manipulations can fundamentally modify the carbon flux dynamics of peatlands, regardless if they are undertaken to prepare the area for commercial use (drainage) or to restore a "natural" state of the ecosystem (rewetting). Anthropogenic use of peatlands usually involves their drainage. The stored carbon can then be oxidized, and a C sink is often turned into a C source. It is estimated that at least 3 billion tons (Parish et al., 2008) carbon dioxide ($CO_2$) are emitted by degraded peatlands per year

globally. This is equivalent to 10 % of the global annual emissions by the combustion of fossil fuels. The rewetting of formerly drained peatlands commonly reduces $CO_2$ emissions drastically and makes the re-establishment of a $CO_2$ sink possible on the long run (Couwenberg, 2009; Wilson et al., 2009; Alm et al., 2007; Vanselow-Algan et al., 2015; Beyer and Höper, 2015; Wilson et al., 2016b; Tuittila et al., 1999). Under water-saturated conditions, however, the anaerobic decomposition of organic matter and thereby the production of the greenhouse gas (GHG) methane ($CH_4$) increases. Land use change of peatlands thus

inheres the potentials to accelerate global warming as well as to mitigate climate change.

For a peatland to act as a $CO_2$ sink, the water level may fluctuate around the surface only to a minor degree. If it is too low, more plant litter is decomposed aerobically than is being produced. If it is too high, plant production often is inhibited, so that e.g. lakes commonly are carbon sources. At water tables near the surface, $CO_2$ emissions are low (respectively negative when C is sequestered); with lowering water tables emisions rise. Couwenberg et al. (2010) found a linear correlation between

$CO_2$ flux ($F_{CO2}$) and water table depth in a meta-analysis of flux data from temperate European peatlands. For sites with mean annual water levels above 40 cm below the surface, $CO_2$ emissions decrease with rising water tables. $CH_4$ emissions are also linked to the water table. At levels deeper than 20 cm below the surface, $CH_4$ emissions are negligible and increase with a rising water table. In case of inundation, diffusive $CH_4$ release is hampered due to the large difference in gas diffusivity of water and air. Moreover, $CH_4$ can be decomposed on its comparably slow way through the water column if enough dissolved

oxygen is present. Two alternative mechanisms for the transport of pedogenic $CH_4$ to the atmosphere are known. $CH_4$ release via bubbles can account for a significant portion of the overall $CH_4$ emissions (Glaser, 2004; Strack et al., 2005; Goodrich et al., 2011). This process is referred to as ebullition and describes the sudden release of gas bubbles that accumulate in the soil pore space until their buoyancy is high enough for them to ascend to the surface. The importance of diffusion and ebullition declines with the presence of vascular plants. The soil and water volume can be bypassed employing plant-mediated





transport through the aerenchymae of vascular plants (Whalen, 2005; Bubier, 1995). Moreover, higher plants also provide labile dissolved organic carbon to the rhizosphere. These easily decomposable carbon compounds can act as a substrate for methanogenic microorganisms. $CH_4$ flux ($F_{CH4}$) dynamics are therefore gravely impacted by vegetation cover and type.

Wilson et al. (2009) investigated the development of $CH_4$ emissions and modeled the course of $CH_4$ emissions for different

land use types following peat extraction. The authors conclude, that by long-term inundation of peatlands formerly used for peat harvesting, the creation of a landscape scale methane hotspot is very possible. Nevertheless, the balance of avoided $CO_2$ emissions by restoration and newly created $CH_4$ emissions results in a net-reduction of the global warming potential (GWP) at the site Wilson et al. (2009) described. When anaerobic conditions prevail after inundation, $CH_4$ production is mainly controlled by the availability of fresh organic matter (Couwenberg, 2009; Lai, 2009; Saarnio et al., 2009) as well as soil and

water temperature (Schrier-Uijl et al., 2010). Hahn-Schöfl et al. (2010) performed a chamber measurement campaign and incubation experiments on a rewetted former grassland fen in the Peene river valley in NE-Germany. The authors describe the formation of an organic sediment from the rotting former vegetation cover. The $CH_4$ production potential linked to the anaerobic decomposition of such a substrate is very high. Tiemeyer et al. (2016) investigated GHG release from 48 grassland sites on drained fens and bogs in Germany. They report high $CH_4$ emissions from relatively nutrient-poor and acidic sites.

Incubation experiments from Hahn-Schöfl et al. (2010) show that bare peat is comparatively inactive. This finding is confirmed by Wilson et al. (2016b) for drained as well as rewetted bare peat surfaces in temperate peatlands. In case of a vegetation-free restored peatland site, the risk of $CH_4$ production depends on which plants are established or colonize the site respectively. Thereby, it is critical how easily decomposable the delivered organic matter is and if plant-mediated methane transport via their aerenchyma (Whalen, 2005) occurs. Furthermore, $CH_4$ production is negatively correlated with the availability of other

electron acceptors like iron or sulfate.

$CH_4$ is an important GHG and a crucial part of the carbon balance of many (especially wetland) ecosystems. While the earliest landscape scale $CH_4$ flux measurements date back to the 1990s (Verma et al., 1992; Zahniser et al., 1995; Suyker et al., 1996), considerable advances in laser absorption spectroscopy (LAS) within the last ten years have led to a wide application of fast LAS-based sensors as part of eddy covariance (EC) setups. Intercomparisons of the available sensors are given by Tuzson

et al. (2010), Detto et al. (2011), Peltola et al. (2013) and Peltola et al. (2014). Due to its low power consumption and thereby its feasibility for remote sites with limited off-grid energy supply, the Licor LI-7700 open-path sensor (McDermitt et al., 2011) has frequently been used in ecosystem $CH_4$ flux studies. Because the measurement cell of such a devices is exposed to the atmosphere, it does not require a pump (reducing the sensor's power requirements) but is also subject to adverse conditions like dust or rain, which can deteriorate the acquired data by contaminating the highly reflective mirrors the sensor relies on.

The development of fast sensors provided the possibility to measure long-term landscape scale $CH_4$ fluxes with the EC technique at high temporal resolution. Ecosystem carbon balances can since be reported more comprehensively. However, to be able to calculate for example annual sums, gaps in the flux time series have to be filled first. Compared to modeling $CO_2$ fluxes, gap-filling of $CH_4$ fluxes is more challenging because the relations between environmental drivers and $CH_4$ flux often appear to be more complex than for $CO_2$. Basic gap-filling methods include for example interpolation between measured values

(Hanis et al., 2013; Dengel et al., 2011) or the use of an average to replace all gaps (Hatala et al., 2012; Mikhaylov et al., 2015).





Simple linear models have also proven to be applicable in certain settings (Alberto et al., 2014; Hanis et al., 2013). A common approach is to fit Arrhenius-type non-linear functions to the flux as a function of various environmental drivers, what has been done for half hourly data (Kroon et al., 2010; Forbrich et al., 2011; Hommeltenberg et al., 2014; Goodrich et al., 2015) as well as for downsampled time series (Suyker et al., 1996; Friborg and Christensen, 2000; Rinne et al., 2007; Long et al., 2010; Wille

et al., 2008; Jackowicz-Korczyński et al., 2010; Parmentier et al., 2011; Brown et al., 2014; Shoemaker et al., 2015; Mikhaylov et al., 2015). However, as stated by Brown et al. (2014), there is evidence that these functional relationships do not necessarily behave monotonically. Artificial neural networks (ANNs) form a category of non-parametric models that have frequently been used to fill gaps in EC $CO_2$ flux time series. Mostly, multilayer perceptrons (MLP) were chosen (Papale and Valentini, 2003; Moffat et al., 2007; Moffat, 2012; Järvi et al., 2012; Pypker et al., 2013; Menzer et al., 2015) while other authors utilized radial

basis function (RBF) networks (Schmidt et al., 2008; Kordowski and Kuttler, 2010; Menzer et al., 2015). For $CH_4$ fluxes, MLP models are described by Dengel et al. (2013), Deshmukh et al. (2014), Knox et al. (2015) and Goodrich et al. (2015) as well as a special kind of RBF network, a generalized regression neural network (GRNN), by Zhu et al. (2013).

In this study, we compare simple linear and more complex neural network modeling approaches to gap-fill half-hourly EC $CH_4$ and $CO_2$ fluxes based on their explanatory power, their number of parameters and their generalization capability. We also

give a structured approach to the choice of architectural properties for ANNs. Additionally, we present a new quality filter for $CH_4$ concentrations measured with LI-7700 (Licor, USA) open-path sensors. By evaluating half-hourly footprint statistics, our data sets from a temperate degraded bog were divided by land use type and gap-filled in order to calculate the annual sums of vertical C exchange between surfaces under contrasting management (drainage and rewetting) and the atmosphere. Based on two years of $CO_2$ and $CH_4$ fluxes, our overriding research questions are: (1) Is gas flux modeling of contrasting surface types

within an EC gas flux time series measured over heterogeneous terrain feasible, and (2) what is the climate impact of peatland land use change from drainage to rewetting in the early phase after ditch-blocking following peat mining in a temperate bog?

## 2   Material and methods

### 2.1   Site description

#### 2.1.1   Geography and land use history

Himmelmoor is a temperate bog that has been degraded heavily by peat mining. The site is located in NW Germany, around 25 km north-east of Hamburg and 3 km west of Quickborn in Schleswig-Holstein (53° 44'23.3" N, 9 °50'55.8" E). The long-term (2000 to 2014) average annual precipitation is 888 mm, measured at a weather station (WMO Station ID: 10146) located 2 km from the peatland center. The mean annual air temperature (2000 – 2014) at this station operated by Deutscher Wetterdienst (DWD) is 9.1 °C. Along with the adjacent grassland-fen of the Bilsbek lowland and the beech-dominated forest

stand Kummerfelder Gehege, Himmelmoor forms a nature reserve according to the EU FFH (Flora Fauna Habitat Directive). Federal law of Schleswig-Holstein protects Himmelmoor as a core area of the local biotope network (Zeltner, 2003). By federal legislation enacted in 1973, human intervention in peatland ecosystems is prohibited in the state of Schleswig-Holstein.





In case of Himmelmoor, however, unlimited lease agreements from 1920 and 1932 (Kreis Pinneberg, 2004) were in effect. An environment protection law from 1993 dictated the expiration of such agreements until 2003. The mining company and the legislators reached a settlement obligating the company to carry out restoration measures while continuing with peat extraction until 2020. Manual peat extraction, undertaken by the local population over centuries to gain fuel, was limited to the bog's outer

margin. Since the mid-19th century, peat-cutting underwent mechanisation and was thereby intensified. Extraction was scaled up in the 1870s, when transport logistics were greatly improved by the construction of a railway track between Quickborn and Altona (today a district of Hamburg). Mining was limitied to peripheral areas until industrial peat extraction began in the 132 ha large (Grube, 2010) central bog area at the beginning of the 20st century (Czerwonka and Czerwonka, 1985). Until 1968, the aim of this operation was the production of peat usable as fuel. Between 1950 and 1968, suitable material was mined

from positions deep in the peat profile. These deep ditches were refilled annually with dug-out peat and still exist in this refilled state today. Locally, these strips are called *Pütten*. Today, fen-type vegetation is covering these strips, which are between 600 m and 700 m long in ENE-WSW direction and 20 m to 50 m wide in NNW-SSE direction. In the late 1960s, the mining company began to target a different, surface-near type of peat to yield a substrate for horticulture. The extraction site is divided into two halves by a NNW-SSE running railroad dam. Areas on the western half have been stepwise taken out of operation by the

local peat plant operator since 2008. The eastern part was still being harvested during the measurement period (1 June 2012 to 31 May 2014), whereas most of the western section was already rewetted, apart from a 90 m wide strip in the northwest (see Figure 1). These areas of opposing water regimes and land use will be referred to as surface class *drained* ($SC_{dra}$) and surface class *rewetted* ($SC_{rew}$) respectively throughout this text. Peat harvesting on the eastern half continued until June 2018, rewetting of smaller sections in this area began, however, already in 2014. Peat extraction is now ceased. Ditches were blocked,

and new peat dams were built to rewet the extraction site, leaving large, shallowly inundated areas underlain by bare peat.

### 2.1.2 Peat formation and stratigraphy

The bog developed in a river valley, which was preformed as a glacial meltwater valley during the Weichselian glaciation in the depressed fringe of a salt dome. Peatland formation from a standing water body began 10.020 ± 100 years BP (Pfeiffer, 1997). Following the deposition of organic lake sediments, a *Phragmites-Carex* fen and a subsequent birch forest formed. Around

8000 years BP, a rising groundwater level led to the extinction of the forest vegetation, to the spread of *Sphagnum spp.* peat mosses and eventually to the development of a bog. Organic-rich silt constitutes the peatland's foundation but is not evenly distributed across the whole area. In the northern part, this layer is at its thickest (up to 2 m). There, the base of the peatland is at its deepest (around 7 m a.s.l.). The formation of the peatland most likely began at this location (Grube et al., 2010). Bog-type peat consisting mostly of *Sphagnum* remnants is (in its original stratigraphic position) today only present in the northern

marginal area, locally called *Knust*, with a thickness of 3.8 m (Grube et al., 2010). Bog peat was completely removed from the central extraction area where fen-type peat is present area-wide with a maximum thickness of 2.2 m. Pfeiffer (1997) divides these peat deposits in 30 – 45 cm *Eriophorum-Betula* peat over 45 – 85 cm birch forest peat over *Phragmites-Carex* peat. Soil properties were altered severely by peat decomposition and subsidence during decades of drainage and peat extraction. Drainage led to compaction and settlement of the peat profile of about 40 % of its original depth. The peatland's ability to





**Figure 1.** Distribution of surface classes *rewetted*, *drained* and *vegetated* in Himmelmoor during the measurement period between 1 June 2012 and 31 May 2014. The map section shows the central extraction area with the EC tower located on the central railroad dam. Grid spacing is 200 m, coordinates refer to UTM zone 32U. The polar histogram in the top left corner displays two years of half-hourly wind direction measurements binned in 2° classes.




self-regulate the water table for optimal peat forming and carbon sequestering conditions is thereby lost. The performed ditch-blocking leads to a strongly oscillating water table over the course of the year in the early years of rewetting (< 5 years, own observations made between 2011 and 2018). Himmelmoor drains into the two local creeks Bilsbek and Pinnau. The peatland mainly receives water from precipitation. Additional minerotrophic water inflow takes place through the *Pütten*, which are distributed regularly across the central bog area. These ditches reach below the peat base and penetrate the mineral ground. They were later refilled with peat but still provide a connection to the aquifer beneath, from which minerotrophic groundwater is supplied.

### 2.1.3 Vegetation

The central, former extraction area is largely vegetation-free. *Sphagnum spp.* peat mosses occur in ditches and along the shores of some rewetted, inundated polders of the former mining area. *Betula pubescens*, *Molinia caerulea*, *Eriophorum angustifolium*, *E. vaginatum*, *Erica tetralix* and brwon mosses (*Odontoschisma spp.*, *Cephalozia spp.*) occur in often isolated patches on drier areas. Fen-type vegetation is common at the *Pütten*. *Betula pubescens*, *Salix spp.*. (presumably *Salix aurita* and *Salix caprea*), *Eriophorum vaginatum*, *Betula pubescens*, *Molinia caerulea*, *Eriophorum angustifolium*, *Calla palustris*, *Typha latifolia*, *Carex spp.*, *Juncus effusus* and *Calamagrostis canescens* occur there.

### 2.2 Instrumentation

Eddy covariance $CH_4$ fluxes were measured using an open-path gas analyzer (LI-7700; Licor, USA) and a 3-D sonic anemometer (R3; Gill, UK) mounted on a tower at 6 m height. Water vapour and $CO_2$ concentrations were determined with an enclosed-path sensor (LI-7200; Licor, USA). Data were recorded on a LI-7550 (Licor, USA) logger at 20 Hz. Additionally, a HMP45 (Vaisala, Finland) temperature and relative humidity probe was mounted on the EC tower and logged with the same device. A second HMP45 was installed together with a NR01 4-component net radiometer (Hukseflux, Netherlands) 70 m southwest of the EC tower on a tripod at 2 m height. These data were logged on a CR-3000 (Campbell Scientific, UK). Another logger of this type was used at the weather station approximately 500 m northeast of the EC tower. The sensors there included a third HMP45 and a tipping bucket rain gauge (R.M. Young, USA). Per depth, redox potentials were determined with three parallel fibreglass probes with platinum sensor tips and recorded on a Hypnos II logger (MVH Consult, Netherlands). The redox probes were installed in a vegetated strip approximately 100 meters west of the EC tower. Water level was measured and logged with a hydrostatic pressure transducer (Mini-Diver; Schlumberger Water Services, USA) around 150 m west-southwest of the EC tower. Rain and long-term temperature data as presented in Figure 2 was taken from a nearby station operated by DWD (WMO-Station ID 10146), which is located east-southeast from the EC tower at approximately 2 km distance. Two years of turbulent flux data were available for analysis from Himmelmoor. The EC setup did not change during that time. The first year from 1 June 2012 to 31 May 2013 is from hereon called Year 1, the second year from 1 June 2013 to 31 May 2014 is called Year 2.





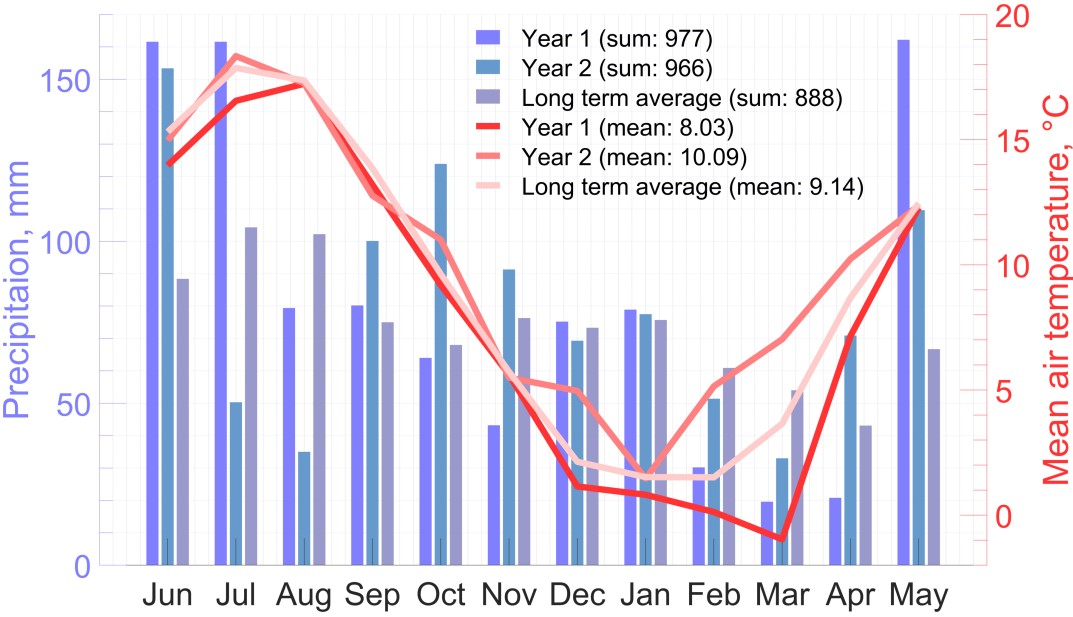

**Figure 2.** Climate diagram of the two investigated years from 1 June 2012 to 31 May 2013 and form 1 June 2013 to 31 May 2014 as measured in Himmelmoor and a 14 year average of a nearby DWD station (WMO-Station ID 10146).

## 2.3   Flux calculation and qualitiy filtering

Turbulent fluxes were computed by applying the EC approach (as for example described in more detail by Aubinet et al.,
2012) using the software EddyPro 5.2.1 (Licor, USA). Raw data processing included (1) an angle of attack correction, i.
e. compensation for flow distortion induced by the anemometer frame (Nakai et al., 2006), (2) coordinate rotation to align
the anemometer x-axis to the current mean streamlines (Kaimal and Finnigan, 1994, double rotation), (3) linear detrending
(Gash and Culf, 1996), (4) time lags compensation, (5) spectral corrections (see below for details) and (6) WPL-correction
to compensate for air density fluctuations due to thermal expansion and water dilution (Burba et al., 2012). High frequency
loss due to path averaging, signal attenuation and finite time response of the instruments was accounted for following Fratini
et al. (2012). Low frequency loss due to finite averaging time and linear raw data detrending was corrected for according
to Moncrieff et al. (2004). More details on the single flux calculation steps are given in Holl et al. (2019a). Thirty minute
fluxes were screened for quality according to the following scheme. To check whether general assumptions necessary for the
application of the EC method were met, atmospheric stability and developed turbulence were analyzed as described by Mauder
and Foken (2004). By this step, fluxes were classified into three groups: MF0, MF1 and MF2, with MF0 denoting data of
highest and MF2 of lowest quality. Due to potentially faulty WPL correction, $CH_4$ and $CO_2$ fluxes of half-hours when sensible
or latent heat flux were flagged with MF2 were discarded. Certain quality flags that were derived from raw data statistics as
described by Vickers and Mahrt (1997) were evaluated. If skewness or kurtosis of vertical wind or sonic temperature were
assigned a hard flag (skewness outside [-2,2], kurtosis outside [1,8]) or if $CH_4$ or $CO_2$ concentration statistics were rated with





a soft flag (skewness outside [-1,1], kurtosis outside [2,5]), trace gas fluxes where discarded. Furthermore, half-hourly fluxes were rejected if the respective 20 Hz concentration time series failed the amplitude resolution test (Vickers and Mahrt, 1997).

Additionally, diagnostic values from the LI-7700 and LI-7200 gas analyzers were used for quality screening. LI-7200 data was omitted when the signal strength indication (AGC) lay above 63. Due to a change in the signal quality definition along with a software upgrade, this rule was modified to discarding data below a value of 75 for data acquired when the sensor was running on firmware version 6.6 and above. With respect to the LI-7700, the sensor's relative signal strenth indication (RSSI) and the heater diagnostics were evaluated. The bottom and top mirror of the gas analyzer's measurement cell can be

heated to counter condensation and frost on the mirrors. The LI-7700 instrument software allows for user-defined thresholds that control the power-on of the heaters. For the bottom heater, a RSSI threshold $RSSI_{th}$, below which the heater is turned on can be adjusted. For the top heater, an ambient temperature offset threshold $T_{a, offset}$ can be defined. This mirror is heated to keep its temperature about $T_{a, offset}$ above ambient temperature. In the present case, $RSSI_{th}$ was set to 20 and $T_{a, offset}$ to 1 °C. The number of samples within one half hour, for which a heater is switched on, is recorded. Accordingly, these diagnostics

(bottom heater on: $BH_{on}$; top heater on: $TH_{on}$) take maximum values of 36000 if a heater is switched on for an entire half hour. Heater diagnostics were investigated closely due to the observation that within an averaging interval, high variation in RSSI was often accompanied by switching events in the 20 Hz heater time series, i.e. if $BH_{on}$ or $TH_{on}$ were neither 0 nor 36000. Moreover, methane concentrations had the tendency to covary with RSSI values if the latter showed large changes, what renders calculated fluxes not trustworthy. In general, the top heater was switched on most of the time whereas switching

events in the $BH_{on}$ time series were more common, which is why we mainly focused on the bottom heater diagnostics for the analysis of this phenomenon. Figure 3 shows the relationship between half-hourly averaged RSSI values, the corresponding relative standard deviation of 20 Hz RSSI ($RSSI_{relStd}$) values and $BH_{on}$. From this graph, we empirically derived a $RSSI_{relStd}$ threshold of 10 % , above which the respective flux records were neglected. Additionally, methane fluxes were discarded if the mean RSSI of the respective averaging interval was below 20.

The next quality screening step addressed the filtering of fluxes related to undesired source areas. We first classified the surface using georeferenced orthoimages of the area. As the surface types we aimed to discriminate were quite large and easily distinguishable on the images, we could draw polygons around the different classes and get the coordinates of their corners. This step was implemented through the Matlab 8.4 Mapping and Image Processing Toolboxes. We defined the surface classes *drained* ($SC_{dra}$), *rewetted* ($SC_{rew}$) and *vegetated* ($SC_{veg}$), the latter of which is contained in the other two surface

types as formerly deep, now refilled and vegetated ditches (*Pütten*). Calculating a 2-D footprint function after Kormann and Meixner (2001) with 1 $m^2$ resolution (see Holl et al., 2019b, for details on footprint model implementation) and summing up the contribution values of all pixels within each of the three surface types, yielded half-hourly class contribution fractions of the different classes to the EC signal ($CC_{rew}$; $CC_{dra}$; $CC_{veg}$). Gas fluxes of half hour intervals when the EC footprint was composed of the railroad dam and areas outside the mining site by more than 70 % were discarded. Fluxes were then filtered for absolute limits. $CH_4$ data outside [-100 1000] nmol $m^{-2}$ $s^{-1}$ and $CO_2$ data outside [-10 10] µmol $m^{-2}$ $s^{-1}$ were neglected.

In case of the $CH_4$ flux time series, outlier removal was addressed furthermore by assessing the frequency distribution of the remaining MF0 data. Values smaller than the bin center of the 1st ($BC_1$) or larger than the 99th percentile's bin center ($BC_{99}$)





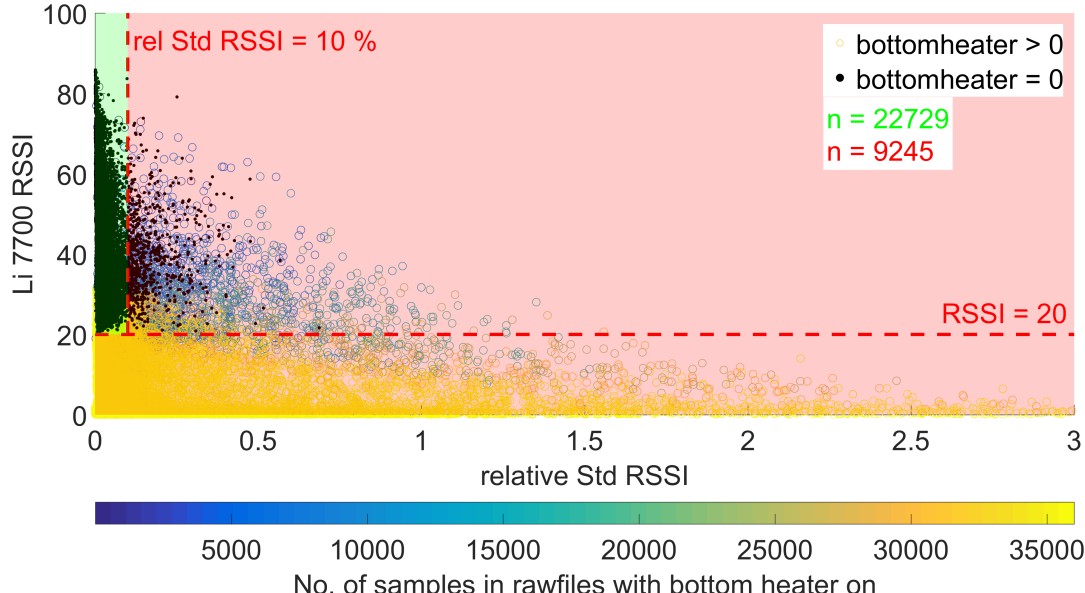

**Figure 3.** Illustration of the empirically derived threshold for the new LI-7700 open path methane analyzer quality filter. This check evaluates raw 20 Hz RSSI statistics to remove erroneous half-hourly methane fluxes. The filter is designed to capture concentration time series that were deteriorated by switching events in a LI-7700 mirror heater. Black data points denote half-hour intervals, during which the heater of the bottom mirror was switched off entirely. Colored points represents half hour intervals, during which switching events (maximum: 36000) in the 20 Hz time series occured.

were omitted. As a last step, $CH_4$ fluxes with random uncertainties calculated with EddyPro after Finkelstein and Sims (2001) larger than 400 nmol $m^{-2}$ $s^{-1}$ were filtered out.

Overall, a large amount of data were removed throughout the course of quality filtering. Of the original 19665 $F_{CH4}$ records 28 % of quality classes MF0 and MF1 were left after filtering. Most fluxes (35 %) were removed because they failed the skewness/kurtosis test. In case of $CO_2$ fluxes, 51 % MF0 and MF1 records of the original 31271 fluxes remained after filtering. The largest amount of data (15 %) were removed because either H or LE of the same half hourly interval were flagged with MF2.

**2.4   Flux modeling**

We applied a two-step gap-filling process to $CO_2$ and $CH_4$ fluxes separately for each of the two available years. We first filled gaps resulting from quality filtering of the measured EC gas fluxes, which represent approximations of the landscape-scale fluxes integrated over the whole ecosystem and thus include areas of contrasting land use types (tower view time series, TVTS). In order to quantify the impact of rewetting on the vertical annual C balances of the peat extraction areas in Himmelmoor, we used EC footprint variables to select fluxes when the EC source area was mainly composed of drained or rewetted surfaces respectively. In particular, we created these surface class time series (SCTS) by selecting all flux values for the class





contributions $CC_{dra}$ and $CC_{rew}$ above a threshold of 70 %. As this data division necessarily resulted in further gaps in the SCTS, we used new sets of models to fill those gaps. We used multilinear regressions (MLRs) and artificial neural networks

(ANNs), in particular multilayer perceptrons (MLPs), to model gas fluxes as a response to measured environmental and EC footprint variables as well as to generated fuzzy logic representations of diurnal and seasonal periodicity. Additionally, we used a model input selection scheme in an effort to exclude redundant and irrelevant variables from the model input matrix. A description of model input selection and model setup is given in Appendix A. Details on data set division by surface class contribution, gap-filling and flux decomposition using a mechanistic approach are given here.

Apart from the class contribution variables, the inputs presented to the selection scheme were the same for TVTS and SCTS modeling. To gap-fill a SCTS, the contributions of the respective opposite surface classes were omitted from the input space. To model the time series representing the rewetted area for instance, $CC_{dra}$ and $CC_{veg,dra}$ (see Table 1) were removed from the input matrix. Also, the surface class of interest was set to the threshold value of 70 % at all flux gaps that resulted from data division. We used this threshold value instead of 100 % class contribution to avoid extrapolating outside the scope of the

training data sets, as the EC footprint was virtually never composed of one surface class only. The measured contributions of the vegetated strips were binned in ten classes and the bin-center of the most frequent class was used to fill gaps in the respective $CC_{veg}$ time series.

We used the selected variables as inputs for MLRs and MLP ensembles with 1000 networks each. To optimize the model parameters, we used only observed data of quality class 0 as targets. Based on the better performance of MLPs compared to

MLRs (see section 3.2), we decided to use MLP models for TVTS as well as SCTS gap-filling. To express model uncertainty, we used the standard deviations of the 1000 single MLPs making up each model ensemble for each half hour. We confirmed normal distribution of the 1000 model fluxes for each half hour by applying a Kolmogorov-Smirnov test at all time steps. To calculate surface class specific annual sums, we included measurement data of quality class 1 back into the SCTS. All quality class 1 values that corresponded to $CC_{dra}$ or $CC_{rew}$ ranging above 70 % were used to replace the modeled SCTS data for the

respective time steps. We filled remaining gaps in the gas flux time series, when no environmental data was available, with a mean diurnal variation (MDV) method (see Falge et al., 2001). If this algorithm encounters a gap, it searches for available values of the same variable in adjacent days at the same hour of day and uses the mean of the found records to fill the gap. At first, a window of ± 1 day around the gap is screened. If not at least one data point is found, the search range is increased in steps of one day until the gap can be filled. We calculated two annual sums for all four SCTS (two gases and two land use

types) from the gap-filled time series. We calculated uncertainty estimates of the annual sums by taking the root of the sum of squared half-hourly uncertainties. For measurements we used the random uncertainty estimate following Finkelstein and Sims (2001), for data modeled with an MLP ensemble the ensemble standard deviation and for data modeled with the MDV method the standard deviation of averaging samples.

In case of the $F_{CO2}$ SCTS, we used a deterministic modeling approach to further decompose the net flux into components related to respiration and photosynthesis. As large parts of Himmelmoor are vegetation-free, we included the class contribution of the vegetated strips (*Pütten*, $SC_{veg}$) to the EC footprint to scale the model terms relating to the respective surface classes (as



**Table 1.** Model inputs used for methane and carbon dioxide flux gap-filling sorted by type. (x: available, -: not available)

| Type | Name | Unit | Quantity symbol | available in... Year 1 | Year 2 |
|---|---|---|---|---|---|
| **Biomet** | Global radiation | W m$^{-2}$ | $R_g$ | x | x |
| | Air temperature | °C | $T_{air}$ | x | x |
| | Outgoing longwave radiation | W m$^{-2}$ | $Lw_{out}$ | x | x |
| | Photosynthetically active radiation | $\mu$mol m$^{-2}$ s$^{-1}$ | PAR | x | x |
| | Air pressure | kPa | $p_{air}$ | x | x |
| | Rate of change in air pressure | kPa/1800 s | $slope_{Pair}$ | x | x |
| | Water vapour pressure deficit | Pa | VPD | x | x |
| | Soil redox potential in 2 cm depth | mV | $Redox_2$ | - | x |
| | Soil redox potential in 5 cm depth | mV | $Redox_5$ | - | x |
| | Soil redox potential in 10 cm depth | mV | $Redox_{10}$ | - | x |
| | Soil redox potential in 20 cm depth | mV | $Redox_{20}$ | - | x |
| | Soil temperature in 2 cm depth | °C | $T_{Soil2}$ | - | x |
| | Soil temperature in 5 cm depth | °C | $T_{Soil5}$ | - | x |
| | Soil temperature in 10 cm depth | °C | $T_{Soil10}$ | - | x |
| | Soil temperature in 20 cm depth | °C | $T_{Soil20}$ | - | x |
| | Soil temperature in 40 cm depth | °C | $T_{Soil40}$ | - | x |
| | Water table below surface | cm | WT | - | x |
| **Fuzzy** | Morning | n.a. | $fuzzy_{mo}$ | x | x |
| | Afternoon | n.a. | $fuzzy_{af}$ | x | x |
| | Evening | n.a. | $fuzzy_{ev}$ | x | x |
| | Night | n.a. | $fuzzy_{ni}$ | x | x |
| | Summer | n.a. | $fuzzy_{su}$ | x | x |
| | Winter | n.a. | $fuzzy_{wi}$ | x | x |
| **Footprint** | Class contribution of rewetted area | n.a. | $CC_{rew}$ | x | x |
| | Class contribution of drained area | n.a. | $CC_{dra}$ | x | x |
| | Class contribution of vegetated area within rewetted part | n.a. | $CC_{veg, rew}$ | x | x |
| | Class contribution of vegetated area within drained part | n.a. | $CC_{veg, dra}$ | x | x |





e.g. in Rößger et al., 2019; Forbrich et al., 2011) in the following way:

$$NEE(CC_{veg}, PAR) = (1 - CC_{veg}) \times TER_{bare} + CC_{veg} \times TER_{veg} - CC_{veg} \times \frac{P_{max} \times \alpha \times PAR}{P_{max} + \alpha \times PAR} \tag{1}$$

where $CC_{veg}$ is the class contribution of the vegetated strips, PAR is photosynthetically active radiation (µmol m$^{-2}$ s$^{-1}$), $TER_{veg}$ and $TER_{bare}$ are ecosystem respirations (µmol m$^{-2}$ s$^{-1}$) of the vegetated strips and the areas covered by bare peat respectively, $P_{max}$ is the maximum photosynthetic rate (µmol m$^{-2}$ s$^{-1}$), and $\alpha$ is the initial quantum yield. Prior to fitting, $CC_{veg}$ and $CC_{bare}$ were rescaled to sum up to 1 so that

$$1 - CC_{veg} = CC_{bare}. \tag{2}$$

The last term of Eq. 1 consists of a rectangular hyperbolic Michaelis-Menten type function to simulate plant photosynthesis (Thornley, 1998; Zheng et al., 2012). This type of light saturation curve has proven to be feasible for modeling plant carbon dioxide fixation driven by radiation. In order for the model to express net $CO_2$ flux, two ecosystem respiration terms were added to the formula; the combined plant and soil respiration $TER_{veg}$ scaled by $CC_{veg}$ and the microbial respiration $TER_{bare}$ taking place in the vegetation-free areas scaled with $CC_{bare}$. This function was fitted to monthly ensembles of all $CO_2$ SCTS, yielding time series of the four model parameters for the drained and rewetted areas for two years. The included scaling of the model terms with the surface class contributions facilitates comparability of the parameter time series among each other and with literature values describing the light response of similar plant communities as found in the vegetated strips of Himmelmoor.

## 3 Results and discussion

### 3.1 Model input selection

Results of our model input selection scheme (see Appendix A) are given in Appendix B. A summarized description of its outcome follows here. Three categories of potential model inputs were presented to the selection scheme: Thirty minute time series of meteorological and soil (Biomet) variables, fuzzy variables representing diurnal and seasonal cycles (following Papale and Valentini, 2003) and footprint variables in the form of surface class contribution estimates. Table 1 gives an overview of the available variables. Note that in Year 1 no soil properties were recorded.

In case of $CH_4$ flux modeling, the selection scheme focused on the footprint contribution of the vegetated strips, VPD and peat temperatures in both years and for both land use types. In Year 1, soil temperatures were indirectly addressed by the inclusion of $Lw_{out}$ (being a measure of surface temperature) and $fuzzy_{su}$ which is correlated strongly to $T_{Soil40}$ (Pearson's correlation coefficient r = 0.9). Diurnal and seasonal cycles were weighted highly by the inclusion of higher frequency fuzzy variables, VPD and $T_{air}$ in both years. In Year 1, $p_{air}$ was included for the rewetted section, stressing the higher likelihood of ebullition events taking place at these inundated areas, which could be facilitated by air pressure variations. The selection of redox potential and water table height time series in Year 2 gives further confidence that our scheme is able to identify physically meaningful driving variables as soil redox conditions are known to be a major limiting factor for methane production in soils.



In case of $CO_2$ flux modeling, all input matrices contain measures for the main driver of photosynthesis which is radiation. PAR and $R_g$ were selected for both land use types in Year 1 and the time-lagged version of PAR for both land use types in Year 2. More emphasis on the impact of plant activity on $CO_2$ fluxes is put by the inclusion of the footprint contribution of the vegetated strips for both years and land use types. A response of $CO_2$ release to respiration is indicated by the selection of peat temperatures, redox potentials and water table height in Year 2 and $Lw_{out}$ in Year 1. $T_{air}$ was selected for all land use types and years and includes both seasonal and diurnal frequency content like the slowly and fast changing fuzzy variables which belong to all final input spaces as well.

## 3.2 Model performance

In order to (I) evaluate the general feasibility of our flux decomposition and modeling method and to (II) compare the performance of MLP and MLR models, we used four approaches. We first compared statistics of MLP and MLR surface class-specific flux models with observed data, which we separated beforehand using an EC footprint model into fluxes relating to either one of the two main surface classes $SC_{dra}$ and $SC_{rew}$. Results are shown in Figures D1 and D2 in the appendix. In all eight cases (two years, two gases, two surface classes), MLPs outperform MLR models with respect to coefficients of determination ($R^2$), Akaike information criterion (AIC) values and root mean squared errors (RMSEs). While MLRs often perform well, there is a tendency for them to overestimate highest and underestimate lowest fluxes in case of $CH_4$ flux modeling leading to S-shaped point clouds around the 1:1 line in the scatter plots given in Figure D1. In case of $CO_2$ models, this tendency of MLRs appears to be less pronounced, whereas there is one case (Year 1, $SC_{dra}$; see Figure D1) where the MLR model explains only 32 % of the measurement data's variability and is therefore evaluated as inept for gap-filling of at least this data set. As a way to evaluate the generalization capability of both model types independently from data used for model optimization, we secondly drove the models which were optimized using Year 1 observations with Year 2 environmental data and compared the results to Year 2 measurements (see Figure D3). Results highlight the applicability of both model types for gas flux time series extrapolation while MLPs again perform superior compared to MLRs in all cases. As expressed in the lower AIC values throughout, the higher model complexity of the MLPs appears to be justified, and the better goodness of fit measures do not seem to be the result of a too tight approximation of the training data. We attribute this result partly to our efforts to reduce the number of hidden layer nodes and the number of independent input variables (i.e. dimensionality reduction of the model input matrices). As a third way to compare model performance as well as to evaluate our method of decomposing gas flux time series obtained with a single EC tower over heterogeneous terrain into surface class-specific time series, we recombined these SCTS by scaling them with their respective contribution to the EC footprint. We calculated the sum of both scaled half-hourly SCTS and compared them to the TVTS measured over heterogeneous terrain at the EC tower. These fluxes include values with mixed surface class contributions also below the threshold of 70 %, which was used to extract target data for SCTS model optimization. Results are shown in Figure D4 in the appendix and again illustrate the better performance of MLPs compared to MLRs. More importantly, the outcome of this circular rescaling experiment demonstrates that after multiple model layers the original measurement data relating to a heterogeneous surface could still be recovered to a reasonable degree.





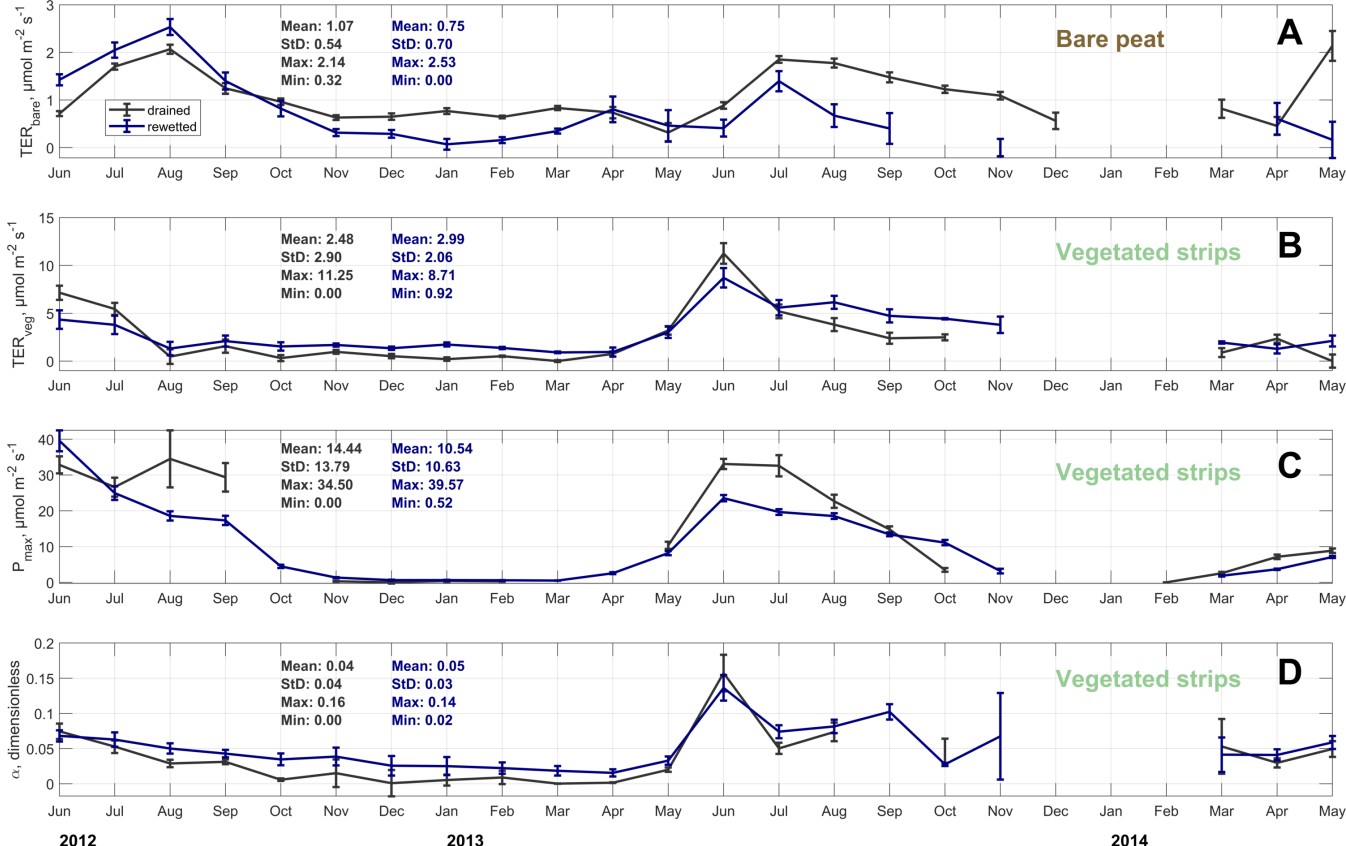

**Figure 4.** Time series of monthly carbon dioxide flux model parameters (ses Eq. 1). Total ecosystem respiration (TER) series are given for the bare (A) and vegetated (B) sections of Himmelmoor. Photosynthesis parameters maximum photosynthesis $P_{max}$ (C) and initial quantum yield $\alpha$ (D) were only determined for areas with vegetation. All parameter time series are given for the rewetted and drained sections of Himmelmoor.

As a fourth method to evaluate the applicability of our land use-specific flux decomposition, we fitted a combined respiration-photosynthesis model (see Eq. 1) to monthly ensembles of the half-hourly $CO_2$ SCTS in order to check if the resultant parameters are reasonable in relation to each other and to literature data. In general, the vegetation period, with its productivity maximum between June and July and its cessation between mid-October and November is well depicted in the seasonal course

5    of the model parameters throughout both years. The parameter courses relating to the vegetated strips of the drained and rewetted areas (Figure 4, panels B – D) develop fairly similar. Distinctions between the drained and rewetted areas are more pronounced with respect to $CO_2$ release from bare peat surfaces (Figure 4, panel A). Ditch-blocking of a rewetted sector close to the EC tower (which therefore made up a large part of the EC footprint) was only performed one year before our measurements started. In summer of 2012 this area therefore was not yet permanently flooded leading to TER$_{bare}$ fluxes exceeding

10   those from the active mining site. From winter 2012/2013 on, inundation of the rewetted bare peat area progressively increased,





resulting in lower $TER_{bare}$ fluxes from the rewetted compared to the drained section. Our $TER_{bare}$ fluxes are in concordance with findings from two studies that were also conducted on the active peat extraction area in Himmelmoor with manual chambers. Vanselow-Algan et al. (2015) report mean annual $CO_2$ emissions from the active mining site of $0.53 \pm 0.05$ µmol m$^{-2}$ s$^{-1}$ with summertime maxima between around 2 and 4 µmol m$^{-2}$ s$^{-1}$. Vybornova et al. (2019) determined similar mean mid-day

fluxes using opaque chambers. For the drained bare peat areas of the extraction site, the latter authors report average fluxes between 0 and 1 µmol m$^{-2}$ s$^{-1}$ throughout the year and a maximum of 3 µmol m$^{-2}$ s$^{-1}$ at a single replicate plot in late August. Vybornova et al. (2019) furthermore measured $CO_2$ release from a rewetted and shallowly inundated bare peat area with opaque floating chambers. The reported mid-day fluxes are generally lower (between 0 and 0.5 µmol m$^{-2}$ s$^{-1}$, summer maximum 1.4 µmol m$^{-2}$ s$^{-1}$) than from the drained extraction site confirming our results. The values of the two mentioned manual

chamber flux studies from Himmelmoor are well within range of the seasonal course of $TER_{bare}$ as it displays for the bare peat areas of $SC_{dra}$ ($1.1 \pm 0.5$ µmol m$^{-2}$ s$^{-1}$) and $SC_{rew}$ ($0.8 \pm 0.7$ µmol m$^{-2}$ s$^{-1}$) alike (see Figure 4, panel A). TER data reported from similar peat extraction sites also agree with our results. Waddington et al. (2002) determined TER fluxes from a bog in Québec (48 °N), Canada, which had been cutover two and three years before dark chamber gas flux measurements. May to August TER averages ranged from 0.8 µmol m$^{-2}$ s$^{-1}$ in a relatively wet to 3 µmol m$^{-2}$ s$^{-1}$ in a comparably dry observed year.

Shurpali et al. (2008) measured TER at an active peat mining site in eastern Finland (62 °N) and determined vegetation period maximum fluxes of 1.3 µmol m$^{-2}$ s$^{-1}$ at the end of August and minima in mid-November of 0.2 µmol m$^{-2}$ s$^{-1}$.

Our combined respiration photosynthesis model (see Eq. 1) includes the relative contributions of the vegetated strips to the EC footprint. Using this method enabled us to compare the extracted model parameter time series (see Figure 4, panels B to D), which can be directly related to plant characteristics, with estimates of these plant species-specific values which have

been determined in other studies for similar plants and plant communities as found in the vegetated strips in Himmelmoor. Additionally, we could distinguish between $CO_2$ release from decomposing bare peat ($TER_{bare}$) and from the vegetated strips ($TER_{veg}$) where respiratory $CO_2$ release also includes autotrophic respiration of plants. In our data set, TER is between twofold and fourfold larger in areas with (Figure 4, panel B) than without (Figure 4, panel A) vegetation. $TER_{veg}$ from the rewetted area is mostly larger than from the drained area. Progressive inundation led to a hydrological connection of $SC_{veg}$ and the flooded

bare peat areas. An increased input of dead plant material as a result of higher water tables might have promoted heterotrophic respiration. Hampered plant productivity due to flooding is also expressed in lower peak values of $P_{max}$ at the vegetated strips of the rewetted site.

Below, we compare our $TER_{veg}$ time series, with literature data from similar plant communities. Using data from chamber measurements, Vanselow-Algan et al. (2015) partitioned NEE from different vegetation communities at the outer edge of

Himmelmoor. These areas were restored three decades ago after being degraded by small-scale manual peat-cutting. Although the plants dominating the vegetated strips of the mining site were not examined by Vanselow-Algan et al. (2015), some of the species investigated by the authors also frequently occur in $SC_{veg}$ being the subject of the present study. The 'purple moor grass' microform in Vanselow-Algan et al. (2015) for example is dominated by *Molinia caerulea*; *Betula pubescens* and *Eriophorum angustifolium* also occur. In summer, TER fluxes form this site were estimated to range above 10 µmol m$^{-2}$ s$^{-1}$

being in the same range like our summer TER peaks from the vegetated strips. Beyer and Höper (2015) report results from





a former north German peat extraction site that was rewetted 30 years prior to their chamber measurement campaign. TER estimates from these authors are available for a site dominated by *Molinia caerulea* (up to 7 µmol m$^{-2}$ s$^{-1}$ in August) and by *Eriophorum angustifolium* (up to 5 µmol m$^{-2}$ s$^{-1}$ in late July). A substantial portion of the SC$_{veg}$ is covered by *Betula pubescens*, *Salix spp.*, *Eriophorum vaginatum*, *Eriophorum angustifolium*, *Typha latifolia*, *Molinia caerulea*, *Carex spp.*, *Juncus effusus*

and *Calamagrostis canescens*. Further combined plant and soil respiration measurements of the species found in SC$_{veg}$ are not present in literature. Nevertheless, properties of plants from the same genera are known. Most reported fluxes, however, describe autotrophic respiration as they were determined on leaf scale and therefore refer to leaf area. Since shrubs and trees can have a leaf area index larger than 1, fluxes refreed to ground surface area could be higher. Measurements of *Betula spp.* dark respiration are given by Patankar et al. (2013) and Gu et al. (2008) (between 1 and 5 µmol m$^{-2}$ s$^{-1}$). Patankar et al. (2013)

also assessed autotrophic respiration of *Salix pulchra* (up to 2 µmol m$^{-2}$ s$^{-1}$), *Eriophorum vaginatum* (up to 3 µmol m$^{-2}$ s$^{-1}$), and *Carex bigelowi* (up to 1 µmol m$^{-2}$ s$^{-1}$). Other *Carex* species are in the same range as shown by Körner (1982) (*Carex curvula* 1 µmol m$^{-2}$ s$^{-1}$) and Murchie and Horton (1997) (*Carex flacca* 1.5 µmol m$^{-2}$ s$^{-1}$). *Salix* summer dark respiration has as well been investigated by Kaipiainen (2009) with *Salix dasyclados* (between 0.8 and 1.2 µmol m$^{-2}$ s$^{-1}$).

Regarding the photosynthesis parameters P$_{max}$ and $\alpha$ in the second model term of Eq. 1, more literature values are available

for comparison. Chamber gas exchange studies of a single birch (*Betula pubescens*) in Himmelmoor during three summer months in 2014 by Lienau (2014) resulted in P$_{max}$ values between 32 and 41 µmol m$^{-2}$ s$^{-1}$. More P$_{max}$ estimates from the same tree species have been reported by Nygren and Kellomäki (1983) (4 to 17 µmol m$^{-2}$ s$^{-1}$) and Hoogesteger and Karlsson (1992). In the latter study, PAR was limited to 800 µmol m$^{-2}$ s$^{-1}$, P$_{max}$ was assessed to be 8 µmol m$^{-2}$ s$^{-1}$. Other evaluations of *Betula spp.* maximum photosynthesis range between 10 and 15 µmol m$^{-2}$ s$^{-1}$ (Patankar et al., 2013; Gu et al., 2008). With P$_{max}$ values commonly around 25 but also above 30 µmol m$^{-2}$ s$^{-1}$ (Chen et al., 2010), *Typha latifolia* is photosynthically more active which is also the case for *Salix spp.* ranging between 16 and 29 µmol m$^{-2}$ s$^{-1}$ (Ögren, 1993). Lab-experiments from Vernay et al. (2016) provide P$_{max}$ estimates of *Molinia caerulea* (7 to 15 µmol m$^{-2}$ s$^{-1}$). For *Juncus effusus* only net photosynthesis values of 6 to 11 µmol m$^{-2}$ s$^{-1}$ (Mann and Wetzel, 1999) have been reported so far. From the North German site investigated

by Beyer and Höper (2015), comparably high P$_{max}$ estimates are reported for *Molinia caerulea* that commonly range between 15 and 30 µmol m$^{-2}$ s$^{-1}$ but also reach values up to 60 µmol m$^{-2}$ s$^{-1}$ in June. The P$_{max}$ parameters given by these authors for *Eriopohorum angustifolium* are also rather large (up to 70, often around 20 µmol m$^{-2}$ s$^{-1}$). Inititial quantum yield estimates of plants also common in in the vegetated strips in Himmelmoor range between 0.02 and 0.08 (Vernay et al., 2016; Nygren and Kellomäki, 1983; Murchie and Horton, 1997; Kaipiainen, 2009; Chen et al., 2010).

In comparison to small-scale measurements from the same and similar sites, we could confirm the credibility of our mechanistic modeling approach for which we used relative class contributions of contrasting surface types to scale single model terms. Moreover, the previously performed division of the TVTS into SCTS apparently yields reasonable flux estimates that can be interpreted in a mechanistic way, increasing our confidence in the applied flux decomposition method.





**Table 2.** Annual sums of half-hourly carbon dioxide ($CO_2$) and methane ($CH_4$) fluxes from the drained and rewetted sections of the peat extraction site in Himmelmoor. $CH_4$ fluxes are expressed as $CO_2$ equivalents ($CO_2e$) using a global warming potential of 34 referring to a 100-year time horizon following Myhre et al. (2013). Year 1: 01 June 2012 to 31 May 2013; Year 2: 01 June 2013 to 31 May 2014

| | | Cumulative gas flux, g m$^{-2}$ a$^{-1}$ | |
| --- | --- | --- | --- |
| | | Surface class *drained* | Surface class *rewetted* |
| $CO_2$ | Year 1 | 988 ± 247 | 887 ± 296 |
| | Year 2 | 974 ± 292 | 567 ± 263 |
| $CH_4$ | Year 1 | 7.2 ± 1.8 | 13.3 ± 1.9 |
| | Year 2 | 12.1 ± 1.4 | 18.3 ± 1.5 |
| $CH_4$-$CO_2e$ | Year 1 | 244 ± 61 | 453 ± 63 |
| | Year 2 | 412 ± 46 | 621 ± 51 |
| total $CO_2e$ | Year 1 | 1232 ± 308 | 1340 ± 359 |
| | Year 2 | 1386 ± 338 | 1188 ± 314 |

### 3.3 Annual greenhouse gas balances

We used the eight (two gases, two land use types, two years) surface-class specific flux time series, which we gap-filled with MLP ensembles, to calculate annual $CO_2$ and $CH_4$ balances for the rewetted and drained sections of Himmelmoor. The results are expressed as molar and mass fluxes (Figure 5) and as release of $CO_2$ equivalents ($CO_2e$, see Table 2). We used a factor of 34 to convert $F_{CH4}$ into $CO_2e$ release. This value is given in the Fifth Assessment Report of the Intergovernmental Panel on Climate Change (IPPC AR5, Myhre et al., 2013), refers to a 100-year time horizon and includes climate–carbon feedbacks.

The impact of rewetting on the development of vertical carbon release is documented with the shown results. Overall, both the rewetted and the mined sections of Himmelmoor were considerable sources of GHGs in both years. Annual $F_{CO2}$ from the restored site undercuts the cumulative $CO_2$ emissions from the drained part of Himmelmoor in both years while this difference increases with time. Annual $CH_4$ release from the wetter surfaces exceeds the cumulative $F_{CH4}$ from the drained mining site in both years while both fluxes rise from Year 1 to Year 2.

In Year 1, $F_{CO2}$ from the rewetted area was already cumulatively lower than from the mining site (20 ± 7 vs. 22 ± 6 mol m$^{-2}$ a$^{-1}$) while the margins of uncertainty largely overlap. In Year 2, the annual $CO_2$ balance from the rewetted site dropped by 35 % increasing the difference to the cumulative mining site flux, which did not change from Year 1 to Year 2, to over 40 % (13 ± 6 vs. 22 ± 7 mol m$^{-2}$ a$^{-1}$). Margins of uncertainty still overlap in Year 2 but less widely. At the end and the beginning of Year 2 (i.e. in summer), the cumulative $F_{CO2}$ curve from SC$_{rew}$ ceases to slope upwards. By reaching these vertexes, the points in time when the rewetted area briefly turns from a $CO_2$ source into a sink are indicated. Nevertheless, on an annual basis the periods when the sink character of SC$_{rew}$ prevails do not compensate for $CO_2$ release during periods of reduced plant activity.

$CH_4$ fluxes from both surface classes rise from Year 1 to Year 2 while the absolute differences between both land use types stays rather constant. The cumulative $CH_4$ flux from SC$_{rew}$ is nearly 90 % higher than from SC$_{dra}$ in Year 1 (0.45 ± 0.11 vs.





$0.83 \pm 0.12$ mol m$^{-2}$) and 50 % higher in Year 2 ($0.76 \pm 0.08$ vs. $1.14 \pm 0.09$ mol m$^{-2}$). Compared to the molar $F_{CO2}$ sums

of both surface classes, cumulative molar $CH_4$ release is a factor of around 30 smaller in Year 1 and about 20 times smaller in

Year 2. The development of both molar GHG emissions over time documents the rising importance of $CH_4$ emissions in the

course of rewetting. Transforming the molar cumulative sums into sums of $CO_2e$ allows for comparability between the two

GHG fluxes with respect to their climate impact. Overall, the rewetted section of Himmelmoor is a larger $CO_2e$ source in the

first observed Year, while the drained section emits more $CO_2e$ in Year 2. $CO_2e$ fluxes at the drained site increase from Year 1

to Year 2, whereas they decline from Year 1 to Year 2 at the rewetted site. The sum of cumulative $F_{CO2}$ and $F_{CH4}$ released

from $SC_{dra}$ are dominated by $F_{CO2}$ in both years. For $SC_{rew}$, $CH_4$-$CO_2e$ emission sums are much smaller than the release of

$CO_2$ in Year 1, whereas $CH_4$-$CO_2e$ fluxes dominate the GWP balance in the second observed year. Although the cumulative

$CH_4$-$CO_2e$ fluxes also increases from Year 1 to Year 2, they mainly dominate the $SC_{rew}$ GWP sum due to a large drop in $F_{CO2}$

from Year 1 to Year 2.

The annual $CO_2$ emissions from the drained parts of $988 \pm 247$ g m$^{-2}$ a$^{-1}$ and $974 \pm 292$ g m$^{-2}$ a$^{-1}$ are higher but in the same

range as previously inferred from chamber data acquired at the mining site in Himmelmoor by Vanselow-Algan et al. (2015)

($730 \pm 67$ g m$^{-2}$ a$^{-1}$) and Vybornova et al. (2019) ($740 \pm 270$ g m$^{-2}$ a$^{-1}$). Moreover, our results are in line with the emission

factors given in the Wetlands Supplement to the 2006 IPCC Guidlines for National Greenhouse Gas Inventories (Hiraishi et al.,

2014) for which Wilson et al. (2016a) published an update. Both publications give an average $CO_2$ release of 1027 g m$^{-2}$ a$^{-1}$

for boreal and temperate peatlands drained for peat extraction. While boreal extraction sites generally appear to emit less

$CO_2$ as reported in a meta-study by Maljanen et al. (2010) ($697 \pm 263$ g m$^{-2}$ a$^{-1}$), Drösler et al. (2008) who analyzed 11

drained peat extraction sites in Europe also report $CO_2$ release of up to 1300 g m$^{-2}$a$^{-1}$. Interesting to note is that in the National

Inventory Report Germany submitted under the United Nations Framework Convention on Climate Change in April of 2019,

$CO_2$ release from drained peat extraction areas are accounted for with a comparably small factor of 587 g m$^{-2}$ a$^{-1}$. Taking

into account, the amount of carbon removed from Himmelmoor by peat extraction ($11000 \pm 1000$ g m$^{-2}$ a$^{-1}$), $CO_2$ emissions

account for less than one tenth of the total carbon loss per year. This value from Vanselow-Algan et al. (2015) is, however,

expressed as $CO_2$ and assumes the instant decomposition of the material after removal. Restored cutover bogs commonly are

$CO_2$ sinks when active peat extraction has been ceased for several decades (Tuittila et al., 1999; Wilson et al., 2016b; Beyer

and Höper, 2015). However, shortly after ditch-blocking, the rewetted section of Himmelmoor still was a considerable $CO_2$

source ($887 \pm 296$ g m$^{-2}$ a$^{-1}$ and $567 \pm 263$ g m$^{-2}$ a$^{-1}$). During regular visits to the area between 2011 and 2019, we observed

that the amplitude of seasonal water table oscillations in a rewetted, formerly mined strip (polder) would decrease with the

time passed after ditch-blocking. We assume that anoxic conditions did not prevail throughout the year in all polders as already

discussed above with respect to $TER_{bare}$ fluxes from these sites. Additionally, ditch-blocking in Himmelmoor went along with

the construction of dams encompassing the newly rewetted polders. Vybornova et al. (2019) showed that shortly after raising

these dams they can be large sources of $CO_2$ with fluxes up to four times larger than from bare peat areas which are drained

for ongoing mining.

       The $CH_4$ flux sums of $13.3 \pm 1.8$ g m$^{-2}$ a$^{-1}$ and $18.3 \pm 1.5$ g m$^{-2}$ a$^{-1}$ from the rewetted sections of Himmelmoor are confirmed

by findings from Beyer and Höper (2015) who report $CH_4$ balances between $16.2 \pm 2.2$ g m$^{-2}$ a$^{-1}$ and $24.2 \pm 5.0$ g m$^{-2}$ a$^{-1}$ from



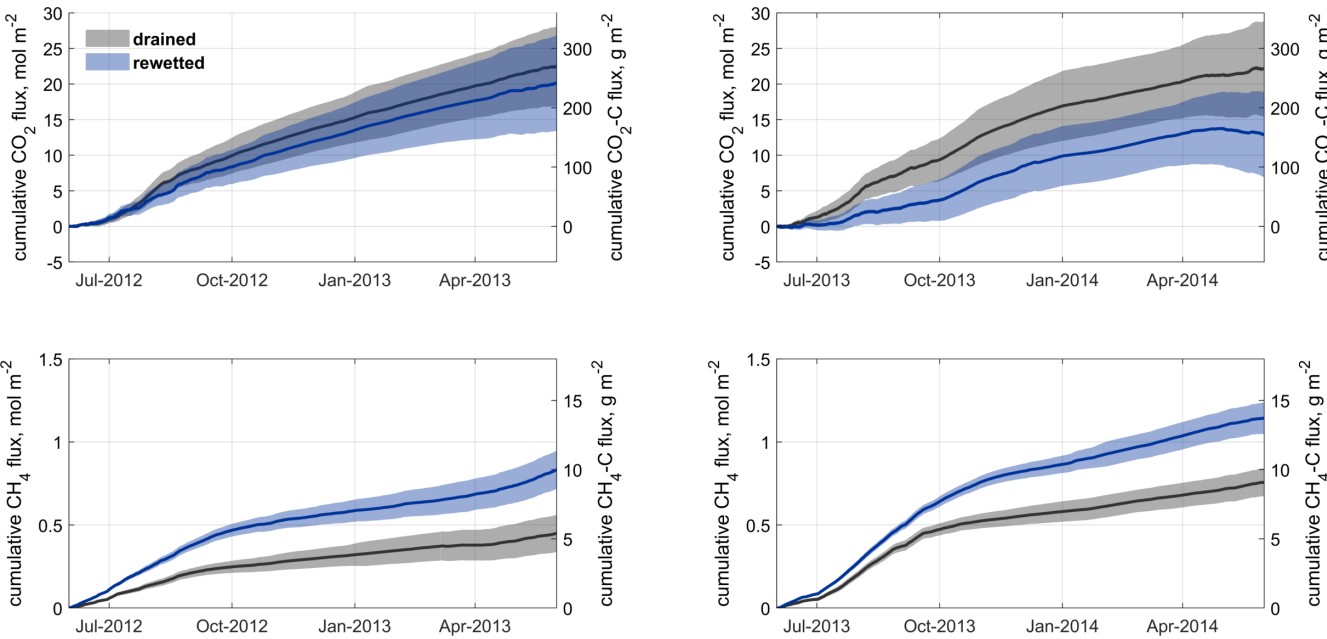

**Figure 5.** Cumulative carbon dioxide ($CO_2$) and methane $CH_4$ fluxes from the drained and rewetted sections of the peat extraction sites in Himmelmoor for both investigated years. Shaded areas represent model uncertainty estimates derived from the standard deviation of artificial neural net ensembles as well as measurement uncertainty estimates. Values are depicted as molar and carbon (C) fluxes.

inundated cutover bogs in northern Germany. Wilson et al. (2016b) report annual $CH_4$ emission sums of $12.0 \pm 2.6$ g m$^{-2}$ a$^{-1}$ from an Irish Atlantic blanket bog that had been rewetted 14 years prior to the investigation. Results from a boreal peat

extraction site, 20 years after mining had been ceased, are given by Tuittila et al. (2000). Although only the growing season has been covered by these authors, the cumulative seasonal $F_{CH4}$ of 1.27 g m$^{-2}$ a$^{-1}$ suggests that $CH_4$ release from boreal peatlands is much lower compared to temperate sites. This circumstance has also been noted by Tiemeyer et al. (2016), who furthermore conclude that IPCC estimates for $CH_4$ release from rewetted bogs, as they are primarily based on data from boreal peatlands, are not representative for temperate regions. In the above mentioned meta-study of Wilson et al. (2016a), temperate

and boreal rewetted peat extraction sites are reported with average emissions of 12.3 g m$^{-2}$ a$^{-1}$. Annual $CH_4$ release from the drained sections of Himmelmoor ($7.2 \pm 1.8$ g m$^{-2}$ a$^{-1}$ and $12.1 \pm 1.3$ g m$^{-2}$ a$^{-1}$) are lower than from the rewetted parts but high compared to IPCC emission factors. Wilson et al. (2016a) give an average release of 3.3 g m$^{-2}$ a$^{-1}$ for drained peat mining sites including a 5 % surface cover of ditches to which the authors assign high $CH_4$ fluxes, as given in Hiraishi et al. (2014) with 10 to 98 g m$^{-2}$ a$^{-1}$. The vegetated strips in Himmelmoor cover around 10 % of the surface and appear to be especially

strong sources of $CH_4$ what we attribute to the high density of vascular, aerenchymatic plants in combination with a supply of nutrient-rich, minerogenic water, which is supplied to these areas from the underlying aquifer.





## 4   Conclusions

As a methodological challenge, we addressed the feasibility of a single EC tower for the estimation of gas flux time series from two surface classes within the EC footprint. Due to the specific setup with (1) the tower position on the border between the surface classes and (2) an adequate measurement height of the EC system, we could attribute fluxes to individual surface classes. This subdivision of EC time series was possible owing to the scale (hundreds of meters) on which surface patterning exists in Himmelmoor. Additionally, the contrast in gas flux dynamics between the different surface classes allowed for a better discriminability between EC fluxes associated to the individual surface types. In situations where flux contrasts are less pronounced and surface classes are more interlaced this method might not be applicable. To be able to estimate gas flux time series of subsections within the EC footprint, rigorous filtering of the original time series is inevitable. Consequently, a considerable amount of measurement data is omitted leading to a relatively high amount of time series gaps. To fill these gaps, we tested multilinear models and artificial neural networks (ANNs) and found that ANNs consistently performed superior. We attribute this fact partly to our data-driven model input selection gravely reducing the number of model parameters. Secondly, we programmatically reduced the number of ANN hidden layer neurons, and thereby furthermore lowered the number of model parameters. Apart from the reduction of the model input space dimensionality, our input selection method also outlined physically sound explanatory frameworks for flux–driver connections. Although it cannot ultimately be appraised if the relevant flux drivers have been captured by our routine, the selection results point to the method's ability to discriminate between weaker and stronger flux–driver relations, which do not necessarily have to be linear. We therefore conclude that when applying empirical models to gap-fill trace gas flux time series, a method for the selection of input variables that takes into account the sensitivity of gas fluxes to model input variability can improve model predictions considerably. When ANNs are used in particular, efforts should be made to set them up in the least convoluted way a sufficient approximation of the target data allows for.

With the estimates of surface class-specific GHG fluxes from two consecutive years, we could address the impact rewetting had on the GHG balance of the peat mining site. Six years after the first polders in Himmelmoor had been rewetted, the area was still a clear GHG source. The two investigated land use types (drainage and rewetting) show distinct $CO_2$ and $CH_4$ flux features. $CO_2$ emissions decrease progressively after rewetting. The release of $CH_4$ increases after rewetting and within the present two year data set also over time. On short timescales, the climate impact of elevated $CH_4$ emissions appears to dominate over the effect of decreasing $CO_2$ release. It is conceivable that Himmelmoor can be transformed into a carbon-accumulating peatland. However, this process will probably take decades to centuries and will take place only when sustained management of the area is employed.



## Appendix A: Model setup and input selection scheme

This section describes the selection of model inputs (see Table 1) using our scoring table approach. Also, the method we used to select properties for the multilayer perceptron (MLP) neural networks is outlined. The MLPs were set up with one hidden layer, tan-sigmoid activation functions, a single output layer node with a linear transfer function and Levenberg-Marquardt

25  backpropagation as supervised learning method. See Papale and Valentini (2003), Dengel et al. (2013), Sarle (1994) for details on MLP architecture. The input data was divided randomly in 70 % training and 30 % validation data. Inputs were re-scaled before training to range between -1 and 1. Training data were used to optimize the network weights and biases for low MSE. Validation data served as inputs independent from training data to check the generalization capability of the model. The model performance in relation to the validation data was used to avoid overfitting by terminating the learning process if for six

30  consecutive iterations the MSE of the validation data did not decrease (early stopping). Instead of using the response of a single MLP, we calculated the ensemble average of multiple networks starting with varying initial weights and different sets of training and validation data each. This method is frequently described in neural network literature (Naftaly et al., 1997; Perrone and Cooper, 1993; Wolpert, 1992; Hashem, 1997; Haykin, 1999) as one type of so called committee machines. To avoid unnecessarily complex network architecture and thereby a higher amount of model parameters we inspected the model performance of committee machines with 100 MLPs each for different numbers of hidden layer nodes (#HLN) between 1 and 20. We expected to find a #HLN optimum at the AIC minimum. However, for different gases and data sets, we encountered two functional forms a relation like this would commonly assume. A parabola-like curve with a clear minimum and an asymptotic function of the form $AIC(\#HLN) = \#HLN^{-1} + a$. We fitted parabolas to the according data sets and assumed the function

vertex as #HLN optimum. In the other cases, we fitted reciprocal functions and differentiated the results. We rounded the first derivatives to the nearest multiple of 10 in case of $CH_4$ and to 100 in case of $CO_2$ flux modeling. We then defined the #HLN optimum to be at the position where the rounded derivative turns zero for the first time. We performed #HLN optimization in each case before applying MLPs for input sensitivity analysis or gap-filling.

We applied a selection procedure aiming for the identification of redundant as well as irrelevant model inputs. This scheme

evaluates the outcome of stepwise MLRs in combination with the analysis of the response of MLPs to differently manipulated versions of the input space. We used methods addressing predictive and causal importance as defined by Sarle (1997). In short, predictive importance measures are those that check the change of model performance when an input is omitted, whereas causal importance measures evaluate the change of a performance function when inputs are manipulated. The latter can be realized by degrading the variability of an input for example by replacing it partly with its average (as in Schmidt et al., 2008;

Hunter et al., 2000). Three categories of potential model inputs were presented to the selection scheme. Thirty minute time series of meteorological and soil (Biomet) variables, fuzzy variables representing diurnal and seasonal cycles (following Papale and Valentini, 2003) and footprint variables in the form of surface class contribution estimates. Table 1 gives an overview of the available variables. Note that in Year 1 no soil properties were recorded. We derived a second set of Biomet variables by estimating the time lag between each Biomet variable and the gas flux time series and subsequently shifting each Biomet

time series by the calculated time lag. We used the lag time within a one-day window for which the absolute cross-correlation





between Biomet and gas flux time series was maximized (Kettunen et al., 1996) to shift the respective Biomet time series. Three data sets were used for sensitivity analysis: Only the original Biomet data, only the lagged data and both. All data sets were extended by fuzzy and footprint data. We applied four methods to estimate the relevance of the individual inputs and combined them via a scoring table. If an input was selected by one method, one point was assigned to it. Inputs with more

points were regarded as more important.

As previously applied by Dengel et al. (2013) for EC flux gap-filling, we used the outcome of a stepwise multilinear regression (MLR) with bidirectional elimination to identify important model inputs. Independent variables that remained in the final model received one point in our scoring table. The calculations were made using the Matlab 8.4 Statistics and Machine Learning Toolbox following Draper and Smith (1998). At each step the p-values of an F-statistic of models with or without

each input were evaluated by comparing them with an enter condition $p_{enter} = 0.05$ and an exit condition $p_{remove} = 0.1$. If inputs currently not in the model had p-values below $p_{enter}$, the one with the lowest value was included into the model until the next step (forward selection). If inputs currently in the model had p-values above $p_{remove}$, the one with the highest value was removed from the model (backward elimination). These steps were repeated until the model could not be improved further by a single step. The initial model contained no inputs.

Following Schmidt et al. (2008), we calculated two similar measures of causal importance from the output of MLP ensembles. The variability of each input variable was manipulated by replacing 50 % and 100 % with its median, while all remaining variables in the input matrix were left unchanged. A MLP ensemble was first trained with the original data and then simulated with the artificial input matrix. The relation of the resultant mean squared errors (MSEs) was calculated and called relative error (RE). This process was repeated 1000 times for all input variables to obtain diverse results for different data divisions. The

resulting values for RE were binned into six classes with centers at 0.8, 0.9, 1.0, 1.1, 1.2 and 1.3. If the latter was the bin with the most counts, one point was assigned to this input variable in the scoring table, meaning that the manipulation of this input vector resulted in a deterioration of the respective MSE of more than 25 % in most cases. This method yielded two measures of causal importance for each input variable, $RE_{50}$ and $RE_{100}$, referring to the two percentages of data being manipulated.

We furthermore analyzed the weights resulting from MLP optimization based on the algorithm of Garson (1991) as presented

in Olden and Jackson (2002). This method interprets the weights of a neural network similar to the coefficients of a linear model. Before calculating the relative importance (RI) of an input, the products of the weights that connect this input with each hidden neuron and the output layer is determined and normalized by the sum of weight products feeding also into the same hidden unit. These so called neuron contributions are summed up and normalized by the sum of all neuron contributions resulting in the RIs of all inputs. We calculated the mean, median and maximum RIs of 1000 MLP runs for all input variables.

We then compiled three lists in which we sorted the inputs in descending order with respect to the determined statistics. The lengths of those lists were afterwards shortened to equal the number of variables that were included in the MLR model that was derived before – only variables with the highest RI statistics stayed in the lists. All inputs that occurred at least in two of three lists received one point in the scoring table, which was completed with this step. We then summed up the scores for all input variables and calculated two score thresholds above which an input was to be selected. One threshold was derived for

the original and the lagged Biomet variables, one for fuzzy and footprint data. We proceeded like this owing to the structure of





the three input data sets. Each Biomet variable occurred in two of three data sets, each fuzzy and footprint variable was part of all data sets, making it more likely for them to reach a high score. We calculated the mean score of the respective variable category and used the next larger integer as a score threshold. The inputs that were selected via the scoring table were fed into a final stepwise MLR removing further apparently irrelevant model inputs. In the last step of the input selection algorithm we
checked if both a variable and its lagged derivative remained in the input matrix. If so, the scores of those two variables were compared, and only the higher scoring variable stayed in the input matrix. In case there was no score difference, the lagged derivative was removed from the input space, whose reduction was hereby finished.

## Appendix B: Model input selection results

To gain first insight into the relations between input variables and landscape-scale gas fluxes (tower view time series, TVTS) as
well as between the input variables among each other, scatter plots were inspected and Pearson's correlation coefficient (r) was determined for each pair. See Table 1 for definitions of the quantity symbols used hereafter. Three Biomet time series correlate with r values of 0.4 or higher with $CH_4$ flux in both years: $Lw_{out}$, $T_{air}$ and $R_g$. In Year 1, this list is extended by VPD and PAR while the highest linear relation exists with $CC_{rew}$ (0.5) and $CC_{veg, rew}$ (0.6). In Year 2, additional connections with r values of 0.4 or higher include soil temperatures $T_{Soil20}$, $T_{Soil2}$ and $T_{Soil40}$. Footprint variables were not as closely related as in Year 1. Nevertheless, $CC_{veg, rew}$ yields again the highest correlation among the footprint variables. Compared to $F_{CH4}$, linear relations between model input variables and $CO_2$ flux are more clear as the only strong connections exist with PAR and $R_g$ (both r = 0.5 in Year 1 and r = 0.6 in Year 2). Regarding linear dependencies between Biomet variables, $R_g$ and PAR (r > 0.9 both years), $T_{air}$ and VPD (r = 0.7 in both years) as well as $T_{air}$ and $Lw_{out}$ (r > 0.9 both years) were highly correlated. In Year 2, soil temperatures
5 were closely connected among each other (r > 0.9) and with $T_{air}$ (r > 0.7). Water table depth was correlated negatively with all redox measurements at different positions in the soil profile, with the largest absolute r of -0.7 for the relation with $Redox_{20}$. WT was also correlated with $T_{Soil20}$ (r = 0.3). The seasonality embedded in soil temperature measurements was reflected by high correlation coefficients with the two low-frequency fuzzy variables fuzzy variable summer ($fuzzy_{su}$) and fuzzy variable winter ($fuzzy_{wi}$). The deeper in the soil profile the temperature measurements were taken, the less amplitude response they show to diurnal variations and the less noisy the relation to the fuzzy data appears to be.

Correlation analysis emphasizes the (not surprising) fact that collinearity does exist in the model input space. In order to avoid overfitting and thereby to increase the predictive power of the applied models, we reduced the input matrices which drive these models using our scoring table approach. Results of this input variables selection are detailed in tables B1 to B4 in the
5 appendix. As a measure to ascertain collinearity reduction, we calculated the condition numbers (Belsley et al., 2005) of the input matrices at successive stages of the selection process as well as for the complete original and time-lagged input series (see figures C1 and C2). Within all 12 data sets, the condition numbers dropped throughout the selection process by at least one order of magnitude denoting a consistent removal of collinear variables from the input space. In all cases, between 30 % and 40 % of the variables presented to the selection scheme were included in the final model input matrices.





10    In the following section, detailed results of our model input selection scheme are shown. The four tables cover two gases and two years. Within each table, results for the two land use types (surface class *drained*, $SC_{dra}$ and surface class *rewetted*, $SC_{rew}$) are shown. See Table 1 for declarations of the used quantity symbols. Only variables reaching a score above the respective score threshold (cf. Appendix A) are included. Variables which were selected in the last step of the scheme and used for gas flux modeling are printed in bold face.

**Table B1.** Result of the model input selection scheme for Year 1 $CO_2$ fluxes. Score threshold for Biomet variables: $SC_{dra}$ (6), $SC_{rew}$ (6). Score threshold for Fuzzy & Footprint variables: $SC_{dra}$ (9), $SC_{rew}$ (10)

| | | Surface class *drained* | | Surface class *rewetted* | |
|---|---|---|---|---|---|
| | | Variable | Score | Variable | Score |
| **Biomet** | | $\mathbf{Lw_{out}}$ | 8 | $\mathbf{Lw_{out}}$ | 8 |
| | | $\mathbf{T_{air}}$ | 8 | $\mathbf{T_{air}}$ | 8 |
| | | $R_g$ | 7 | $\mathbf{R_g}$ | 8 |
| | | **PAR** | 7 | $Lw_{out}$, lagged | 8 |
| | | $Lw_{out}$, lagged | 7 | PAR | 7 |
| | | $R_g$, lagged | 7 | VPD, lagged | 6 |
| | | $T_{air}$, lagged | 7 | $T_{air}$, lagged | 6 |
| **Fuzzy & Footprint** | | $\mathbf{CC_{veg,\,dra}}$ | 12 | $\mathbf{CC_{veg,\,rew}}$ | 12 |
| | | $\mathbf{fuzzy_{su}}$ | 12 | $\mathbf{fuzzy_{wi}}$ | 12 |
| | | $\mathbf{fuzzy_{wi}}$ | 12 | $\mathbf{fuzzy_{af}}$ | 11 |
| | | $\mathbf{fuzzy_{af}}$ | 9 | $\mathbf{fuzzy_{ni}}$ | 11 |
| | | $\mathbf{fuzzy_{ev}}$ | 9 | $\mathbf{fuzzy_{mo}}$ | 10 |





**Table B2.** Result of the model input selection scheme for Year 2 $CO_2$ fluxes. Score threshold for Biomet variables: $SC_{dra}$ (6), $SC_{rew}$ (6). Score threshold for Fuzzy & Footprint variables: $SC_{dra}$ (9), $SC_{rew}$ (9)

|  | Drained | | Rewetted | |
|---|---|---|---|---|
|  | Variable | Score | Variable | Score |
| **Biomet** | **$T_{air}$** | 8 | **$T_{air}$** | 8 |
|  | **PAR, lagged** | 8 | **$T_{Soil2}$** | 8 |
|  | $T_{Soil2}$ | 7 | **$T_{Soil5}$** | 8 |
|  | **$Redox_5$** | 7 | **$T_{Soil10}$** | 8 |
|  | $T_{air}$, lagged | 7 | **PAR, lagged** | 8 |
|  | **$T_{Soil20}$, lagged** | 7 | **$T_{Soil40}$** | 7 |
|  | **$Redox_2$, lagged** | 7 | $T_{Soil40}$, lagged | 7 |
|  | $T_{Soil5}$ | 6 | $T_{Soil2}$, lagged | 7 |
|  | $T_{Soil10}$ | 6 | $T_{Soil5}$, lagged | 7 |
|  | $Redox_{10}$ | 6 | $T_{Soil10}$, lagged | 7 |
|  | $Redox_{20}$ | 6 | **$T_{Soil20}$, lagged** | 7 |
|  | **WT, lagged** | 6 | $T_{Soil20}$ | 6 |
|  | $T_{Soil2}$, lagged | 6 | **$Redox_2$** | 6 |
|  | **$T_{Soil10}$, lagged** | 6 | **$Redox_{10}$** | 6 |
|  | $Redox_5$, lagged | 6 | $T_{air}$, lagged | 6 |
|  | **$Redox_{10}$, lagged** | 6 | $Redox_{20}$, lagged | 6 |
|  | $Redox_{20}$, lagged | 6 | | |
| **Fuzzy & Footprint** | **$CC_{veg, dra}$** | 12 | **$fuzzy_{su}$** | 11 |
|  | **$fuzzy_{wi}$** | 9 | **$fuzzy_{wi}$** | 11 |
|  | **$fuzzy_{af}$** | 9 | **$CC_{veg, rew}$** | 9 |
|  | **$fuzzy_{ev}$** | 9 | **$fuzzy_{af}$** | 9 |
|  | **$fuzzy_{ni}$** | 9 | **$fuzzy_{ev}$** | 9 |
|  | **$fuzzy_{su}$** | 9 | **$fuzzy_{ni}$** | 9 |





**Table B3.** Result of the model input selection scheme for Year 1 $CH_4$ fluxes. Score threshold for Biomet variables: $SC_{dra}$ (6), $SC_{rew}$ (6). Score threshold for Fuzzy & Footprint variables: $SC_{dra}$ (8), $SC_{rew}$ (8)

| | | Drained | | Rewetted | |
|---|---|---|---|---|---|
| | Variable | Score | Variable | Score | |
| **Biomet** | **VPD** | 8 | **VPD** | 8 | |
| | **$T_{air}$, lagged** | 8 | $Lw_{out}$, lagged | 8 | |
| | **$Lw_{out}$** | 7 | $T_{air}$, lagged | 8 | |
| | $T_{air}$ | 7 | **$Lw_{out}$** | 7 | |
| | VPD, lagged | 7 | **$p_{air}$** | 7 | |
| | $Lw_{out}$, lagged | 6 | **$T_{air}$** | 7 | |
| **Fuzzy & Footprint** | **$CC_{veg, dra}$** | 12 | **$CC_{veg, rew}$** | 12 | |
| | **$fuzzy_{su}$** | 12 | **$fuzzy_{su}$** | 12 | |
| | **$fuzzy_{af}$** | 10 | **$fuzzy_{af}$** | 11 | |
| | $fuzzy_{mo}$ | 8 | **$fuzzy_{mo}$** | 8 | |
| | **$fuzzy_{wi}$** | 8 | | | |





**Table B4.** Result of the model input selection scheme for Year 2 $CH_4$ fluxes. Score threshold for Biomet variables: $SC_{dra}$ (6), $SC_{rew}$ (7). Score threshold for Fuzzy & Footprint variables: $SC_{dra}$ (8), $SC_{rew}$ (9)

| | | Drained | | Rewetted | |
| --- | --- | --- | --- | --- | --- |
| | Variable | Score | Variable | Score |
| **Biomet** | **VPD** | 8 | **VPD** | 8 |
| | **$T_{Soil40}$** | 8 | WT | 8 |
| | **$T_{Soil5}$** | 8 | **$T_{Soil5}$** | 8 |
| | $Redox_{10}$ | 8 | **$T_{Soil20}$** | 8 |
| | $T_{Soil40}$, lagged | 8 | **$Redox_{10}$** | 8 |
| | $T_{Soil10}$, lagged | 8 | **$Redox_{20}$** | 8 |
| | **$Redox_5$, lagged** | 8 | WT, lagged | 8 |
| | **$Lw_{out}$** | 7 | **$Redox_2$, lagged** | 8 |
| | $T_{Soil2}$ | 7 | **$T_{Soil40}$** | 7 |
| | **$T_{Soil10}$** | 7 | $T_{Soil2}$ | 7 |
| | **$Redox_2$** | 7 | $T_{Soil40}$, lagged | 7 |
| | **$T_{Soil2}$, lagged** | 7 | **$T_{Soil2}$, lagged** | 7 |
| | $T_{Soil5}$, lagged | 7 | $T_{Soil5}$, lagged | 7 |
| | $T_{Soil20}$, lagged | 7 | $T_{Soil5}$, lagged | 7 |
| | $Redox_2$, lagged | 7 | $Redox_{20}$, lagged | 7 |
| | $Redox_{10}$, lagged | 7 | | |
| | $T_{air}$, lagged | 6 | | |
| | **$T_{Soil20}$** | 6 | | |
| | **$Redox_{20}$** | 6 | | |
| | VPD, lagged | 6 | | |
| | **WT, lagged** | 6 | | |
| | $Redox_{20}$, lagged | 6 | | |
| **Fuzzy & Footprint** | **$CC_{veg, dra}$** | 12 | **$CC_{veg, rew}$** | 12 |
| | **$fuzzy_{su}$** | 10 | **$fuzzy_{wi}$** | 12 |
| | **$fuzzy_{mo}$** | 9 | **$fuzzy_{mo}$** | 9 |
| | **$fuzzy_{af}$** | 8 | **$fuzzy_{su}$** | 9 |



**Appendix C:  Effect of dimension reduction of model input space on matrix condition**

In this section, matrix condition numbers of the differently manipulated versions of input variable combinations that were fed
into the input selection scheme (first three groups from the left in the plots below) and condition numbers of matrices at the
two final stages of the selection scheme (last two groups from the left in the plots below) are given. Lower condition numbers
denote a smaller degree of linear dependencies within different variables in a matrix. See Appendix A for details on the input
selection method. Three data sets were modeled for each gas flux time series per year: The originally measured EC fluxes
(tower view) representing landscape-scale integrated fluxes and the extracted time series, using EC footprint modeling, which
relate to areas under different land use (drainage and rewetting) are shown.

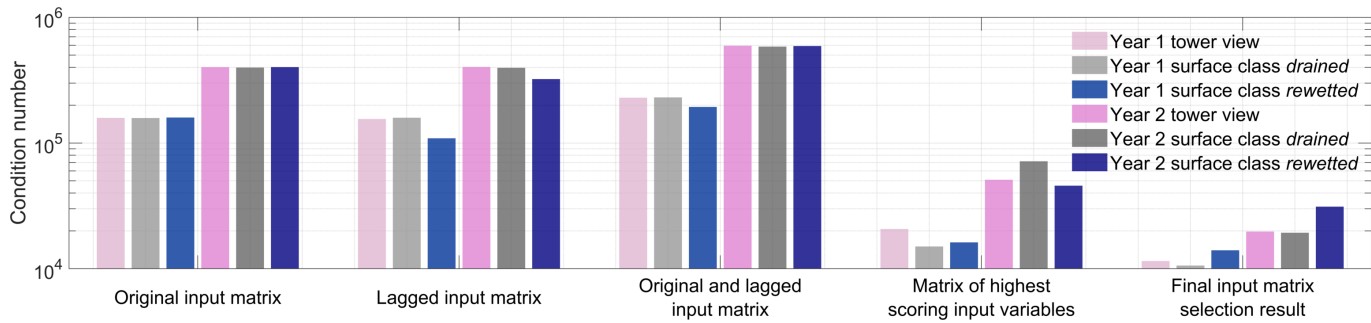

**Figure C1.** Matrix condition numbers of input combinations which were fed into the $CH_4$ flux model input selection scheme (first three
groups from the left) and condition numbers of matrices at the two final stages of the selection scheme (last two groups from the left).

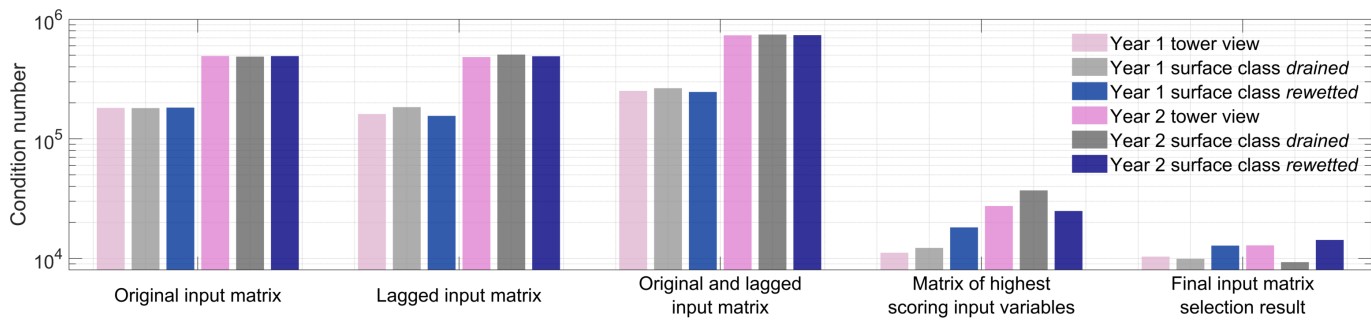

**Figure C2.** Matrix condition numbers of input combinations which were fed into the $CO_2$ flux model input selection scheme (first three
groups from the left) and condition numbers of matrices at the two final stages of the selection scheme (last two groups from the left).



## Appendix D: Comparative validation of multilayer perceptron and multilinear regression models

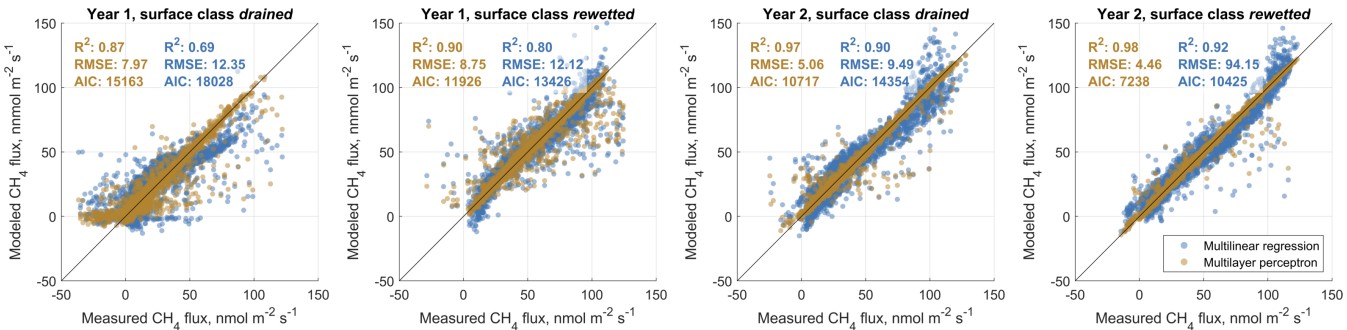

**Figure D1.** Comparative validation of surface class-specific multilinear regression and artificial neural network (in particular multilayer perceptron) methane ($CH_4$) flux models for both investigated years. Multilayer perceptrons appear to be superior with respect to the coefficient of determination ($R^2$), the Akaike information criterion (AIC) and the root mean squared error (RMSE) in all cases. Moreover, multilinear regression models tend to be S-shaped and therefore overestimate high and low measured fluxes.

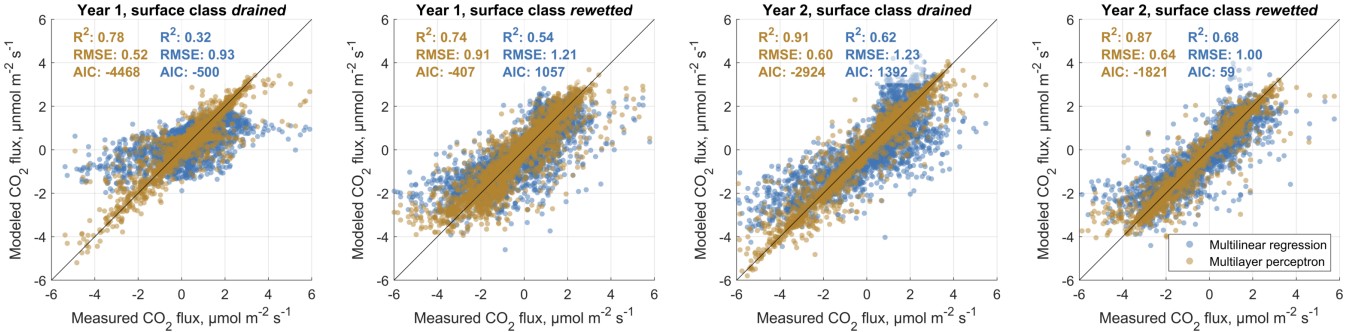

**Figure D2.** Comparative validation of surface class-specific multilinear regression and artificial neural network (in particular multilayer perceptron) carbon dioxide ($CO_2$) flux models for both investigated years. Multilayer perceptrons appear to be superior with respect to the coefficient of determination ($R^2$), the Akaike information criterion (AIC) and the root mean squared error (RMSE) in all cases.

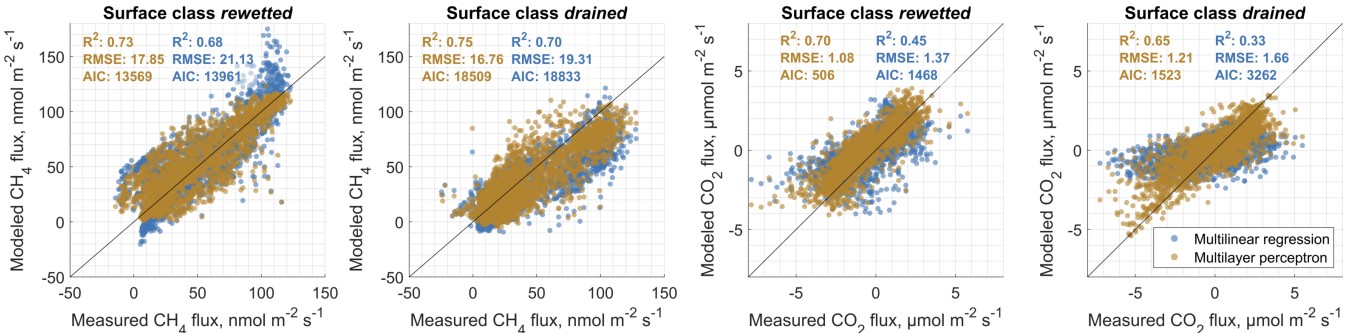

**Figure D3.** Comparative validation of surface class-specific multilinear regression and artificial neural network (in particular multilayer perceptron) carbon dioxide ($CO_2$) and methane ($CH_4$) flux models. For this depiction, we drove models that were optimized using Year 1 measurement data as targets with Year 2 environmental data and compared the results to measured Year 2 gas fluxes. This type of comparison enables an evaluation of the developed models with observed data which is completely independent from model optimization. Therefore, good agreement cannot be attributed to models which are overfit to the provided target data. The results of this investigation substantiate the notion that multilayer perceptrons provide more reliable estimates of gas fluxes as they are superior to multilinear models with respect to the coefficient of determination ($R^2$), the Akaike information criterion (AIC) and the root mean squared error (RMSE) in all cases.

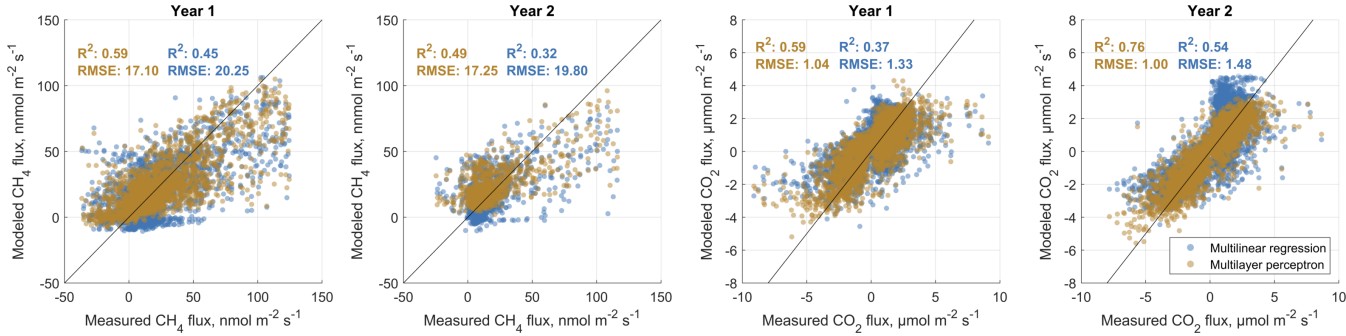

**Figure D4.** Comparative validation of multilinear regression and artificial neural network (in particular multilayer perceptron) carbon dioxide ($CO_2$) and methane ($CH_4$) flux models. For this analysis, both surface class-specific flux models were combined using the footprint contribution of the respective classes yielding an estimate of the landscape-integrated signal recorded at the eddy covariance tower. These derived 'tower view' gas flux time series are compared to the actually observed data at the eddy covariance tower. In all cases, the multilayer perceptron models' performances surpass the attainment of multilinear regression models with respect to the coefficient of determination ($R^2$) and the root mean squared error (RMSE). Results also highlight the general feasibility of our approach to decompose an eddy covariance (EC) time series recorded over heterogeneous terrain into contributions of differently functioning landscape units within the EC footprint area.







*Author contributions.* LK and EMP conceptualized and administered the planning of the research activity and acquired the funds for it. DH and LK conducted the field work. DH conducted literature research, analyzed the data, created visualizations and wrote the original draft. LK, EMP and DH reviewed and edited the original draft.

*Competing interests.* No competing interests are present

*Acknowledgements.* None of the research projects carried out in Himmelmoor would have been possible without the fantastic, continuous support of the peat plant operators. We want to thank the manager of the peat plant, Klaus-Dieter Czerwonka, his wife Monika, their son Hans and Hans Müller for their always reliable, hands-on approach to the diverse challenges that come with the installation and operation of measurement equipment in hard to access areas. For the technical planning and setting up of the measurement equipment we cordially
thank Peter Schreiber and Christian Wille. For supporting the fieldwork in Himmelmoor, we thank Norman Rüggen, Ben Runkle, Olga Vybornova, Adrian Heger, Tim Pfau, Tom Huber, Ben Kreitner, Laure Hoeppli, Anastasia Tatarinova, Zoé Rehder and Oliver Kaufmann. We thank Peter Klink for helping with literature research. This work was supported through the Cluster of Excellence CliSAP (EXC177), Universität Hamburg, funded through the German Research Foundation (DFG).





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
