# Peer review of "Comparison of eddy covariance CO2 and CH4 fluxes from mined and recently rewetted sections in a NW German cutover bog"

_Biogeosciences, 2019_

## Referee Comment (RC1) · Anonymous Referee #1 · 21 Jan 2020

This manuscript reports carbon dioxide and methane fluxes for the period June 2012 to May 2014. Using a combination of a single eddy covariance tower, footprint modeling, and manual spatial cover classification using remotely sensed images, the authors distinguish, separately gap-fill, and quantify annual sums for, both actively mined and recently rewetted peat sections. The authors find that rewetting increases methane and decreases carbon dioxide emissions, but those effects manifest themselves much more strongly in the second year after rewetting, indicating lags.

Overall the paper is clearly written but could be much shorter. The strongest aspects of the study are the comprehensive scholarship and the clarity of the methods. For

example, there is a clear description of eddy covariance data processing for methane, which seems to have been considered with great care, and is an active area of research in the flux community (e.g., European RINGO initiative, perhaps should be linked more specifically). The exploration of gap-filling approaches is also a nice addition, though I think it takes up too much of the paper overall, given that is not the primary focus of the study (not even in the title). There are however some issues with the paper that I think need to be addressed which I outline below.

Major Comments

Soil conditions In year 2 the authors report a substantial amount of soil data being recorded, including temperature, redox, and water table height. These in turn are included via their variable selection procedure in the predictive models of methane flux. Unfortunately, these data are not presented to the reader at all. This is disappointing as the focus of the paper implied by the title is the difference in fluxes between the two cover types, and soil conditions are likely the mechanism underlying those differences by year 2. I would encourage the authors to explore visualizations of those soil data in the paper, perhaps by substituting it for some of the discussion of either the machine learning or the $CO_2$ discussion.

Flux Partitioning Why was the net ecosystem exchange flux partitioning done at the monthly timestep? Can this not be performed at half-hour timesteps in EddyPro? I assume this was done intentionally but the justification is not clear.

Synthesis Literature summaries in the introduction and the discussion need to avoid listing. I am referring to the carbon dioxide flux sections, whereas the methane section is better synthesized (I especially like the comparison to IPCC values). The comparisons made in the results to other studies might be better tabulated. If they are noted in the main text, they should be synthesized better.

Machine Learning In Appendices A and B, the authors outline the machine learning approach used (artificial neural networks). Can the authors justify why they used a

single data split as opposed to a k-fold cross validation approach, which tends to give a more stable performance evaluation? Using the alternative year as a "test" set for generalizability is interesting.

Can the authors also comment on whether gaps were artificially created during validation, or whether the data splits were performed randomly on all observations?

Style I personally enjoyed the descriptive style of the writing, but it is unfortunately much to verbose for a modern readership. I would encourage the authors to mercilessly edit to reduce text. They might be surprised how much shorter the paper is if written in a more declarative style. An example:

"We used a factor of 34 to convert FCH4 into CO2e release. This value is given in the Fifth Assessment Report of the Intergovernmental Panel on Climate Change (IPPC AR5, Myhre et al., 2013), refers to a 100-year time horizon and includes climate–carbon feedbacks"

. . .could be shortened to:

"We used a CH4 global warming potential of 34 kg CO2-eq (IPCC AR5, Myhre et al., 2013), which assumes a 100-year time horizon and includes climate-carbon feedbacks."

Or:

"Nevertheless, on an annual basis the periods when the sink character of SCrew prevails do not compensate for CO2 release during periods of reduced plant activity."

"Nevertheless, annually integrated ecosystem respiration outweighs photosynthesis in SCrew."

Minor Comments

Page 2 Line 9: Perhaps "land-use or climate change" rather than "men"

Line 14: "of carbon dioxide"

Line 20: "inheres the potentials" is ambiguous phrasing

Line 29: perhaps "oxidized" rather than "decomposed"

Page 3 Line 3: perhaps "strongly" rather than "gravely"

Lines 4-20: This is a nice minireview, but could be stronger if structured more systematically, or if the points could be linked more, to sound less like a list.

Page 7 Line 14: "brown"

Lines 20-: Can you briefly justify the variable positions of these sensors? How representative is the water-level sensor of the general footprint?

Figure 1: Can you please add some more points for the other sensor installations.

Page 8 Figure 2: This figure can be more useful to visualize how each true calendar year deviates from the long-term average if it just showed the full timeseries in one series (June 2012-May 2014). The problem currently is that it is difficult to visually integrate the deviations from the mean.

Line 8: Is WPL strictly a correction?

Page 18 Table: Acres are not SI units. Please report in m2, hectares (ha), or km2

Line 33: I think the values in parentheses should be reversed given the order of the sentence.

---

## Referee Comment (RC2) · Anonymous Referee #2 · 23 Jan 2020

**General comments**

This manuscript tested the use of single EC tower to estimate  $CO_2$  and  $CH_4$  fluxes from different land surface area (drained and rewetted) in a mined bog in Northwest Germany by partitioning the sources of signals using footprint statistics. It is an interesting paper from both technical perspective and management perspective.

The manuscript in general is well-written. The authors paid special attention to the footprint analysis, which is very good as here we bend the rules for applying eddy covariance technique. And the gap-filling procedure, the model input selection and the comparison of model performance are clearly explained, although it even seems a bit

too technical considering the main topic is the comparison of EC CO2 and CH4 fluxes from different surface types of a restored bog. But it is a matter of style.

One thing not much mentioned in the paper is the information about the processes and controls which I generally have interest in. How did the environmental variables affect the fluxes under different water regime? how important was temperature control, water table and photosynthesis at different time scales in these ecosystems?

I do like the comparison of TER between bare peat and vegetated strips as shown in fig.4. And I would also like to see similar comparison for CH4. Vegetated strips are in close proximity of EC tower from both rewetted and drained section. By merely looking at the frequency of wind directions, the most frequent wind direction are apparently from the vegetation stripes. "The vegetated strips in Himmelmoor cover around 10% of the surface and appear to be especially strong sources of CH4...", as stated in the paper, it further proved the importance of vegetation on CH4 flux. Thus it would be more interesting and useful to quantify the CH4 flux from vegetation and bare peat separately, rather than solely reporting the annual balance of the mixture. In addition, the section about the vegetation is currently very simple. It would be nice if the authors can provide more information on the vegetation as the EC tower is located just near by. For example , there are tree species like *Betula pubescens*. How tall are they? Can there be flow distortion since the EC mast is not very high (2m)?

"In summer of 2012 this area therefore was not yet permanently flooded ...From winter 2012/2013 on, inundation of the rewetted bare peat area progressively increased,..." It would be nice to show the time series of water table level during the study period. How was the dynamics and intensity of the inundation with time? I also wonder if the vegetated area was changing during the study period due to the progressive inundation. It was shown by a previous study that the fractal dimension of the vegetation area has the most importance in explaining the variation of fluxes in a restored wetland (Matthes et al., 2014). The authors have done a nice job reporting the annual greenhouse gas balances and comparing them to other studies. But we should also be careful here, about the reasons behind those numbers. As I can see from the paper, vegetation and progressive inundation have substantial contribution to the results. Imagine if the EC tower is moved somewhere else with higher (or lower) fraction of vegetation in its footprint, or if the measurements are conducted one year before (or later), are we expecting to get similar results from the drained and rewetted sections? Some sensitivity tests would help to show the reliability of results.

In the end, I do like to see a bit of advices concerning the management of the peat-extraction fields. For examples, is it advisable to rewet the field in terms of climate impact? What are the pros and cons of having large patches of vegetation during rewetting? Should we aim to regulate the water level during the rewetting?

**Specific comments**

Abstract

Page 1, Line 18: The numbers in CO2 fluxes from rewetted and drained section are wrong. Rewetted section should have lower CO2 emission as stated in the manuscript. Page 1, Line 20: It is not useful to compare the difference in CH4 to the difference in CO2 in an absolute term. Surely CO2 is larger in the magnitude.

Introduction

Page 2, Line 27-29: Please provide references.

Page 3, Line 1: What do you mean by "higher plants"?

Material and methods

Page 5, Line 19: So the drained section had also some area rewetted during the study period? Please specify what you meant here.

Page 7, Line 15: Salix spp..

Page 11, Line 1: Maybe replace "and" with "or"

Page 11,Line 13:"...70 % at all flux gaps that resulted from data division". What does

СЗ

that mean?

Page 11, Line 15-17: Why not using median? Maybe a probability density function plot would justify your method for gapfilling  $CC_{veg}$ .

Results and discussion

Page 14, Line 12-20: Was the ANN model prediction compared with the testing subsets as it should be? As written in Appendix A, "70 % training and 30 % validation data", it seems there is no testing subset of data.

Page 20, Line 20: There are many more recent studies on that topic. e.g. "Impact of water table level on annual carbon and greenhouse gas balances of a restored peat extraction area", Jarveoja et al., 2016, Biogeosciences.

Conclusions

Page 21, Line 16-17: "The release of CH4 increases after rewetting and within the present two year data set also over time." This sentence does not read very well.

Page 21, Line 17-18: This statement does not correspond to the current results. CO2 decreased from 887 to 567 g m-2yr-1 while CO2e of CH4 increased from 453 to 621 g m-2yr-1 in the rewetted section from year 1 to year 2. Otherwise, comparing the rewetted to the drained section, CO2 dropped from 974 to 567 g m-2yr-1 and CO2e of CH4 increased from 412 to 621 g m-2yr-1 in year 2. Either way it showed the reduction of CO2 emission was more prominent than the increase of CH4 emission during rewetting.

**Technical comments**

1) Maybe some of the figures D1-4 can be moved to the main text as they validated the modelling and the flux decomposition method.

2) The results on the cumulative fluxes were repeatedly presented in multiple units (g  $m^{-2}a^{-1}$  in Table 2, mol  $m^{-2}$ , CO2-C g  $m^{-2}$  and CH4-C g  $m^{-2}$  in Figure 5). Maybe Table 2 and figure 5 can be combined instead.

**References**

Matthes, Jaclyn Hatala and Sturtevant, Cove and Verfaillie, Joseph and Knox, Sara and Baldocchi, Dennis: Parsing the variability in CH4 flux at a spatially heterogeneous wetland: Integrating multiple eddy covariance towers with high-resolution flux footprint analysis, JOURNAL OF GEOPHYSICAL RESEARCH-BIOGEOSCIENCES, 119, 7, 1322-1339, 2014.

---

## Author Response (AR1)

Author reply to Referee comments from **Anonymous Referee # 1** from 21 January 2020 (https://doi.org/10.5194/bg-2019-432-RC1) on:

**"Comparison of eddy covariance CO$_2$ and CH$_4$ fluxes from mined and recently rewetted sections in a NW German cutover bog" by David Holl et al.**

Reviewer comments (RC)
Author comments (AC)
Mentioned line numbers refer to the originally submitted manuscript
Manuscript changes (MC)

This manuscript reports carbon dioxide and methane fluxes for the period June 2012 to May 2014. Using a combination of a single eddy covariance tower, footprint modeling, and manual spatial cover classification using remotely sensed images, the authors distinguish, separately gap-fill, and quantify annual sums for, both actively mined and recently rewetted peat sections. The authors find that rewetting increases methane and decreases carbon dioxide emissions, but those effects manifest themselves much more strongly in the second year after rewetting, indicating lags. Overall the paper is clearly written but could be much shorter. The strongest aspects of the study are the comprehensive scholarship and the clarity of the methods. For example, there is a clear description of eddy covariance data processing for methane, which seems to have been considered with great care, and is an active area of research in the flux community (e.g., European RINGO initiative, perhaps should be linked more specifically). The exploration of gap-filling approaches is also a nice addition, though I think it takes up too much of the paper overall, given that is not the primary focus of the study (not even in the title). There are however some issues with the paper that I think need to be addressed which I outline below.

**Major Comments**

Soil conditions
In year 2 the authors report a substantial amount of soil data being recorded, including temperature, redox, and water table height. These in turn are included via their variable selection procedure in the predictive models of methane flux. Unfortunately, these data are not presented to the reader at all. This is disappointing as the focus of the paper implied by the title is the difference in fluxes between the two cover types, and soil conditions are likely the mechanism underlying those differences by year 2. I would encourage the authors to explore visualizations of those soil data in the paper, perhaps by substituting it for some of the discussion of either the machine learning or the CO2 discussion.

As mentioned by the referee, our modeling approach does include an identification procedure for likely flux drivers. We also present a short (section 3.1) and extended (Appendix B) discussion on how these drivers can explain flux variability in a mechanistic way. Most likely because our data set was measured over heterogeneous terrain, we did not find a comparably simple flux—driver relation (e. g. with soil temperature or water table) which explained the observed flux variability to a sufficient degree so it could be used to gap-fill our high-frequency data in order to calculate annual flux balances. Due to the complexity of the flux data set, we decided to use a more complex modeling approach. Nevertheless, we agree that it is necessary to depict the site conditions more clearly so a reader can more easily compare to conditions at similar sites and grasp our data set quicker and more comprehensively. We therefore implemented a new modeling approach representing methane flux as a function of soil temperature and water table and explored the results in a new figure and an additional paragraph in section 3.1.

To further investigate the relation between CH₄ flux and the identified likely drivers, we fitted an exponential model of water table and soil temperature (in 40 cm depth) to the CH₄ fluxes from the rewetted section (see Figure XX1). With the exponential dependence of CH₄ flux on soil temperature, a fair amount ($R^2$ = 0.55) of the flux variability can be explained while the added water table term allows for the optimized temperature-$F_{CH4}$ curve to take two distinct paths above and below an approximate water table threshold of 20 cm below the surface (see Figure XX1, panel A). Half-hourly flux variability is, however, substantial due to the heterogeneity of the site's surface and other confounding factors like for example the above-mentioned air pressure variations and is comparably better explained by our neural network models (see Figure XX1, panel B).

[Figure]

*Figure XX1*      *Panel A: Observed Half-hourly methane (CH4) fluxes from the rewetted section of Himmelmoor modeled as an exponential function of soil temperature in 40 cm depth and water table (FCH4(Tsoil/WT)). Monthly and daily flux and temperature averages are also given. Panel B: Comparison of a more complex artificial neural network (ANN) model with the exponential model from Panel A. Although methane flux variability can be explained by the exponential model to a reasonable degree, the level of complexity in flux—driver relations appears to be represented considerably better by the ANN.*

**Flux Partitioning**

Why was the net ecosystem exchange flux partitioning done at the monthly timestep? Can this not be performed at half-hour timesteps in EddyPro? I assume this was done intentionally but the justification is not clear.

No, flux partitioning is not a capability EddyPro provides. We applied our own model (Eq. 1, page 13), and used half-hourly fluxes to optimize the parameters of Eq. 1 (as stated in line 2, page 15). Due to the surface heterogeneity of our site, the independent parameters we chose differ from those used in other common approaches (e.g. Reichstein et al., 2005). We included half-hourly footprint characteristics and radiation but not temperature as drivers of net ecosystem exchange (NEE). Therefore, total ecosystem respiration (TER) in our model is a parameter. To depict the seasonal course of the parameter time series, we optimized one set of parameters for each month of the two-year data set (Fig. 4, page 15). The resolution of the ecosystem respiration time series is therefore limited to monthly steps. In an effort to yield NEE models, which are less likely to be overfitted, we reduced the number of model inputs and parameters and chose monthly, rather than for example daily, flux ensembles for optimization. We were not able to achieve a higher temporal resolution of TER confidently. We could, however, have calculated gross primary production (GPP) for half-hourly intervals using thirty minute radiation measurements

and the two parameters Pmax and α (panels C and D in Fig. 4, page 15). Due to the focus of this paper (annual NEE balances and gap-filling), we decided to omit this step at this point. Instead, we compared the determined photosynthesis parameters, which directly relate to plant characteristics, to literature values of plants that also occur at our site in order to examine the credibility of our land use-specific gap-filling models (as stated on page 15, line 1).

**Synthesis Literature summaries** in the introduction and the discussion need to avoid listing. I am referring to the carbon dioxide flux sections, whereas the methane section is better synthesized (I especially like the comparison to IPCC values). The comparisons made in the results to other studies might be better tabulated. If they are noted in the main text, they should be synthesized better.
We agree, literature synthesis was a bit wordy in the running text. We created three new tables. One for the introduction that summarizes methane gap-filling methods and two in the results section (3.2) where we compare our $CO_2$ model parameters to literature values. We propose to move the tables to the appendix.

We replaced page 3, line 34 to page 4, line 12 with:
...the relations between environmental drivers and CH4 flux often appear to be more complex than for $CO_2$. An overview of methods applied in EC literature is given in Table A 1. Basic gap-filling methods include for example interpolation between measured values or the use of an average to replace all gaps. Simple linear models have also proven to be applicable in certain settings. A common approach is to fit Arrhenius-type non-linear functions to the flux as a function of various environmental drivers. However, as stated by Brown et al. (2014), there is evidence that these functional relationships do not necessarily behave monotonically. Artificial neural networks (ANNs) form a category of non-parametric models that have frequently been used to fill gaps in EC CO2 flux time series. Mostly, multilayer perceptrons (MLP) were chosen (Papale and Valentini, 2003; Moffat et al., 2007; Moffat, 2012; Järvi et al., 2012; Pypker et al., 2013; Menzer et al., 2015). Most recent literature on $CH_4$ flux gap-filling assess MLP models to be the most robust. MLPs are recommended within the processing for the pan-European Integrated Carbon Observation System (ICOS) by Nemitz et al. (2018) and for the new methane component of FLUXNET and the Global Carbon Project's efforts better constrain the global methane budget respectively (Knox et al., 2019).

**Appendix A: Gap-filling methods from literature**

Table A1. Overview of methods applied in literature to gap-fill eddy covariance methane flux time series.

| Method | | References |
|---|---|---|
| Interpolation | | Hanis et al. (2013); Dengel et al. (2011) |
| Averaging | | Hatala et al. (2012); Mikhaylov et al. (2015) |
| Arrhenius-type non-linear functions | half-hourly | Kroon et al. (2010); Forbrich et al. (2011); Hommeltenberg et al. (2014); Goodrich et al. (2015) |
| | downsampled | Suyker et al. (1996); Friborg and Christensen (2000); Rinne et al. (2007); Long et al. (2010); Wille et al. (2008); Jackowicz-Korczyński et al. (2010); Parmentier et al. (2011); Brown et al. (2014); Shoemaker et al. (2015); Mikhaylov et al. (2015) |
| Look-up tables | | Pypker et al. (2013); Hommeltenberg et al. (2014); Bhattacharyya et al. (2014) |
| Mean diurnal variation | | Dengel et al. (2011); Jha et al. (2014) |
| Marginal distribution sampling | | Alberto et al. (2014); Shoemaker et al. (2015) |
| Machine learning | Artificial neural networks | Dengel et al. (2013); Deshmukh et al. (2014); Knox et al. (2015); Goodrich et al. (2015); Nemitz et al. (2018); Knox et al. (2019); Kim et al. (2019) |
| | Support vector machines | Kim et al. (2019) |
| | Random forest | Kim et al. (2019) |

I replaced page 15, line 1 to page 17, line 9 with:

[revised manuscript text omitted]

**Machine Learning In Appendices A and B**, the authors outline the machine learning approach used (artificial neural networks). Can the authors justify why they used a single data split as opposed to a k-fold cross validation approach, which tends to give a more stable performance evaluation? Using the alternative year as a "test" set for generalizability is interesting. Can the authors also comment on whether gaps were artificially created during validation. or whether the data splits were performed randomly on all observations?

I assume the referee refers to the division of target data (fluxes) into training and validation sets in the course of network optimization. To my understanding, we actually did use a simple 2-fold cross validation by dividing the data set into two groups. Due to the large number of fluxes (especially in case of methane) that we discarded during quality filtering, a division into more groups would have resulted in a lower number of fluxes per group, impairing network training. As we used ensemble averages of 1000 networks and therefore performed network optimization 1000 times, we also (randomly) divided the target data differently each time and in my opinion sufficiently counteracted effects of overfitting by this proceeding.

**Style**

I personally enjoyed the descriptive style of the writing, but it is unfortunately much too verbose for a modern readership. I would encourage the authors to mercilessly edit to reduce text. They might be surprised how much shorter the paper is if written in a more declarative style.

Thank you for the feedback. I agree and cut down on verbosity by replacing large parts of running text with tables as suggested in previous comments from Referee #1.

An example:

"We used a factor of 34 to convert FCH4 into CO2e release. This value is given in the Fifth Assessment Report of the Intergovernmental Panel on Climate Change (IPPC AR5, Myhre et al., 2013), refers to a 100-year time horizon and includes climate–carbon feedbacks"

...could be shortened to:

"We used a CH4 global warming potential of 34 kg CO2-eq (IPCC AR5, Myhre et al., 2013), which assumes a 100-year time horizon and includes climate-carbon feedbacks."

Although I understand the referee's general notion, I do not think this is a good example. The edited sentence is shorter mostly because the abbreviation "IPCC AR5" is not explained. In my opinion, also commonly used and widely known abbreviations should be explained when they first occur for consistency.

Or:

"Nevertheless, on an annual basis the periods when the sink character of SCrew prevails do not compensate for CO2 release during periods of reduced plant activity."

"Nevertheless, annually integrated ecosystem respiration outweighs photosynthesis in SCrew."

I agree, sentence replaced.

**Minor Comments**

Line 9: Perhaps "land-use or climate change" rather than "men"

Changed

Line 14: "of carbon dioxide"

Changed

Line 20: "inheres the potentials" is ambiguous phrasing

"Inheres" replaced with "has".

Line 29: perhaps "oxidized" rather than "decomposed"

"decomposed" replaced with "converted to $CO_2$".

Line 3: perhaps "strongly" rather than "gravely"

Changed

Lines 4-20: This is a nice minireview, but could be stronger if structured more systematically, or if the points could be linked more, to sound less like a list.

No change made. The questions are: What is known from literature about the development of methane emissions after peatland rewetting? What is to be expected for a largely vegetation-free site like Himmelmoor? To me the structure is systematic and the points are linked. I am not sure what to change.

Page 7 Line 14: "brown"

Changed

Lines 20-: Can you briefly justify the variable positions of these sensors? How representative is the water-level sensor of the general footprint?

I extended the description of sensor positions:

"A second HMP45 was installed together with a NR01 4-component net radiometer (Hukseflux, Netherlands) 70 m southwest of the EC tower on a tripod at 2~m height. The radiation sensors were not mounted on the EC tower because the field of view of the downward-facing sensors would have covered the peat dam and therefore not be representative for a dominant surface type at the site. These additional HMP45 and NR01 data were logged on a CR-3000 (Campbell Scientific, UK). Another logger of this type was used at the weather station which was taken over from a previous project and for data consistency was left at a position approximately 500~m north of the EC tower.

The water level within the footprint is highly variable as the surface consists of drained and rewetted sections. Our single sensor is representative for the rewetted bare peat strip to the southwest of the EC tower making up a large part of the EC footprint when wind comes from southwesterly directions.

Figure 1: Can you please add some more points for the other sensor installations.
I updated the map with locations of all measurement systems

[Figure]

Figure 2: This figure can be more useful to visualize how each true calendar year deviates from the long-term average if it just showed the full timeseries in one series (June 2012-May 2014). The problem currently is that it is difficult to visually integrate the deviations from the mean.
Ok. Figure restyled.

[Figure]

Line 8: Is WPL strictly a correction?
It is true that there is a discussion about this topic as compensation for air density fluctuations can also be seen as part of the eddy covariance method itself and not as a post-processing step and therefore does not qualify as a correction. On the other hand, the term WPL-correction is still widely used in the community, likely for historical reasons. I did not change this terminology.

Page 18 Table: Acres are not SI units. Please report in m2, hectares (ha), or km2
"$a^{-1}$" stands for "per annum/year". We do report area in $m^2$. No change made.

Line 33: I think the values in parentheses should be reversed given the order of the sentence.
True, thank you for the hint, order was reversed.

Author reply to Referee comments from **Anonymous Referee # 2** from 23 January 2020 (https://doi.org/10.5194/bg-2019-432-RC2) on:

**"Comparison of eddy covariance CO$_2$ and CH$_4$ fluxes from mined and recently rewetted sections in a NW German cutover bog" by David Holl et al.**

Reviewer comments (RC)
Author comments (AC)
Mentioned line numbers refer to the originally submitted manuscript
Manuscript changes (MC)

**General comments**

This manuscript tested the use of single EC tower to estimate CO2 and CH4 fluxes from different land surface area (drained and rewetted) in a mined bog in Northwest Germany by partitioning the sources of signals using footprint statistics. It is an interesting paper from both technical perspective and management perspective.

The manuscript in general is well-written. The authors paid special attention to the footprint analysis, which is very good as here we bend the rules for applying eddy covariance technique. And the gap-filling procedure, the model input selection and the comparison of model performance are clearly explained, although it even seems a bit too technical considering the main topic is the comparison of EC CO2 and CH4 fluxes from different surface types of a restored bog. But it is a matter of style.

We see the Referee's point; the model description is a bit lengthy. It would be a possibility to move Appendix A to a supplementary document as it contains only a detailed description of the methods. If I understand Copernicus' rules correctly, all other appendices cannot to moved to the Supplements as they contain results and interpretations. We want to ask the editor for his opinion on moving Appendix A to the supplements. We would argue against removing the algorithm description section entirely as for reproducibility and transparency the methods should be documented somewhere.

One thing not much mentioned in the paper is the information about the processes and controls which I generally have interest in. How did the environmental variables affect the fluxes under different water regime? how important was temperature control, water table and photosynthesis at different time scales in these ecosystems?

The focus of the paper is the impact of land use change on the annual balances of methane and carbon dioxide fluxes. Presumably due to the heterogeneous surface of the site, we did not find simple flux—driver relations. We addressed this complexity by using models that allow for the characterization of discontinuous and non-linear responses of spatially integrated fluxes (as measured with the EC system) to environmental drivers and source area variations. We, however, realize that it is necessary to depict the site conditions more clearly so a reader can more easily compare to conditions at similar sites and grasp our data set quicker and more comprehensively. We therefore implemented a new modeling approach representing methane flux as a function of soil temperature and water table and explored the results in a new figure and an additional paragraph in section 3.1.

To further investigate the relation between CH$_4$ flux and the identified likely drivers, we fitted an exponential model of water table and soil temperature (in 40 cm depth) to the CH$_4$ fluxes from the rewetted section (see Figure XX1). With the exponential dependence of CH$_4$ flux on soil temperature, a

fair amount ($R^2$ = 0.55) of the flux variability can be explained while the added water table term allows for the optimized temperature-$F_{CH4}$ curve to take two distinct paths above and below an approximate water table threshold of 20 cm below the surface (see Figure XX1, panel A). Half-hourly flux variability is, however, substantial due to the heterogeneity of the site's surface and other confounding factors like for example the above-mentioned air pressure variations and is comparably better explained by our neural network models (see Figure XX1, panel B).

[Figure]

Figure XX1        Panel A: Half-hourly methane (CH4) flux from the rewetted section of Himmelmoor as an exponential function of soil temperature in 40 cm depth and water table (FCH4(Tsoil/WT)). Monthly and daily flux and temperature averages are also given. Panel B: Comparison of a more complex artificial neural network (ANN) model with the exponential model from Panel A. Although methane flux variability can be explained by the exponential model to a reasonable degree, the level of complexity in flux—driver relations appears to be represented considerably better by the ANN.

I do like the comparison of TER between bare peat and vegetated strips as shown in fig.4. And I would also like to see similar comparison for CH$_4$. Vegetated strips are in close proximity of EC tower from both rewetted and drained section. By merely looking at the frequency of wind directions, the most frequent wind direction are apparently from the vegetation stripes. "The vegetated strips in Himmelmoor cover around 10% of the surface and appear to be especially strong sources of CH$_4$...", as stated in the paper, it further proved the importance of vegetation on CH$_4$ flux. Thus it would be more interesting and useful to quantify the CH4 flux from vegetation and bare peat separately, rather than solely reporting the annual balance of the mixture.

Agreed, separate methane flux models for the bare and vegetated areas would be desirable. With the limited amount of available data after quality filtering, we were, however, not able to confidently decompose the measured fluxes of the mixtures into time series only referring to the vegetated strips like it was possible in case of carbon dioxide fluxes. For methane, "only" a decomposition into the rewetted and drained section (addressing the main topic of the study, the impact of land use change on GHG balances) was possible for us to accomplish.

We, however, agree that a depiction of the impact of the contribution of the vegetated strips to the EC footprint on methane fluxes should still be added to our manuscript and therefore added new figures. The impact of footprint variability on half-hourly EC flux variability is included in a new Figure XX1 and more specifically addressed in a new Figure XX2. In the latter figure, comparisons of fluxes when the

vegetation contribution to the EC footprint was below and above 20 % respectively are shown as boxplots. Systematic distinctions between those two groups (and also the low number of available measurements) are illustrated. Please note the added paragraph shown in response to the referee's comment to Page 20, Line 20 which also refers to the new Figure XX2.

[Figure]

Figure XX2    *Dependence of methane fluxes on wind direction and eddy covariance (EC) source area composition, in particular the contribution of the vegetated strips, in summer (A) and winter (B). Data of both investigated years are shown. The  EC tower was placed on a railroad dam dividing the area into an actively (East) and formerly (West) mined section, which had been rewetted prior to measurements. In general, methane emissions from the rewetted section were higher than from the drained section and fluxes when the EC footprint was composed of more than 20 % vegetated areas was significantly (Two-sample Kolmogorow-Smirnow test, p < 0.01) higher than from vegetation-free areas, both in summer (C) and winter (D).*

In addition, the section about the vegetation is currently very simple. It would be nice if the authors can provide more information on the vegetation as the EC tower is located just near by. For example , there are tree species like Betula pubescens. How tall are they? Can there be flow distortion since the EC mast is not very high (2m)?

Actually, the EC sensors were mounted at 6 meters height (see page 7, line 20).  Tree height in the vegetated strips was up to 2 m. We took differences in roughness length in different wind sectors into account within our footprint model by statistically determining individual roughness length estimates for 2° wind direction bins. We believe that we addressed variations in roughness length sufficiently thorough.  We added a sentence after page 9, line 18 for clarity.

Variations in vegetation height and thereby roughness length in different wind sectors were addressed by statistically determining roughness length estimates separately for 2° wind sectors prior to evaluating the footprint model (see Holl2019b for details).

We amended the vegetation description on page 7, line 17 with:

B. pubescens and Salix spp. reached heights of up to 2 m and a combined estimated surface cover within the vegetated strips of up to 10 %.

"In summer of 2012 this area therefore was not yet permanently flooded …From winter 2012/2013 on, inundation of the rewetted bare peat area progressively increased,…" It would be nice to show the time series of water table level during the study period. How was the dynamics and intensity of the inundation with time?

Unfortunately, water table data are not available for the whole study period, only for the second year (starting in June 2013). We added a new Figure XX1 to illustrate the annual course of water table depth in conjunction with soil temperature and methane fluxes.

I also wonder if the vegetated area was changing during the study period due to the progressive inundation. It was shown by a previous study that the fractal dimension of the vegetation area has the most importance in explaining the variation of fluxes in a restored wetland (Matthes et al., 2014).

Vegetation cover was well established in the vegetated strips (former deep ditches, refilled with peat in the late 1960s) and not the direct result of rewetting. The rewetted former mining areas were largely vegetation-free during our investigation. The contrast between vegetated and unvegetated areas as well as the fraction of vegetated areas in the landscape stayed virtually constant.

The authors have done a nice job reporting the annual greenhouse gas balances and comparing them to other studies. But we should also be careful here, about the reasons behind those numbers. As I can see from the paper, vegetation and progressive inundation have substantial contribution to the results. Imagine if the EC tower is moved somewhere else with higher (or lower) fraction of vegetation in its footprint, or if the measurements are conducted one year before (or later), are we expecting to get similar results from the drained and rewetted sections? Some sensitivity tests would help to show the reliability of results.

Yes, as our data were acquired during a major transition of the site from active peat mining to restoration and I would expect gas flux characteristics to further change in the future. Although, inter-annual variability and systematic shifts in processes cannot be separated precisely with our two-year data set, it also can be seen as a document of shifting conditions and ecosystem response mechanisms. We are confident that the position of the measurement equipment was chosen adequately in order to describe the gas flux dynamics of the site as a whole. The fraction of the vegetated strips within the EC footprint resembles the actual proportion of this surface class within the investigation area. So, yes, moving the tower would potentially change the results. We think our data set is suitable for the purpose of characterizing landscape-scale integrated gas fluxes from the surface class mixture prevalent in Himmelmoor. We added a new Figure XX3 as evidence for this notion.

[Figure]

*Figure XX3        Frequency distribution of relative half-hourly contributions of vegetated strips to EC footprint area in both investigated years (Year 1: 01 June 2012 to 31 May 2013; Year 2: 01 June 2013 to 31 May 2014 ). For comparison, the vegetated strips' areal fraction within the investigation area is shown, documenting that the measurement system was set up at an adequate position in the landscape in order to represent its spatial proportion of surface classes.*

In the end, I do like to see a bit of advices concerning the management of the peat-extraction fields. For examples, is it advisable to rewet the field in terms of climate impact? What are the pros and cons of having large patches of vegetation during rewetting? Should we aim to regulate the water level during the rewetting?

We added a section discussing the rewetting measures which have been taken at Himmelmoor.

**3.4 Implications of rewetting measures for the re-establishment of a mire ecosystem in Himmelmoor**

In general, the initialization of peat accumulation by Sphagnum mosses is inevitable (Joosten, 1992; Pfadenhauer and Klötzli,1996; Gaudig, 2002) for the purpose of re-establishing a degraded peatland's natural ecosystem functions. Two (somewhat untypical) features of Himmelmoor need to be considered when evaluating the success of the implemented rewetting measures in terms of mire re-establishment and climate change mitigation: (1) The fact that large vegetation-free areas have been inundated shallowly and (2) that fen-type plants have established at the only vegetated areas which had been taken out of use in the late 1960s. We found peak $CH_4$ emissions from the vascular plant-dominated areas (see Figure 10) and also attribute this fact causally to the presence of fen-type vegetation. Vascular plants provide an effective transport pathway through their gas-conducting tissue as well as root exudates which form an easily decomposable substrate for soil microbes (Kerdchoechuen, 2005; Neue et al., 1996; Bhullar et al., 2014). Because a water table above the surface instead of close to but below the surface has been established at the bare peat areas, the creation of floating vegetation mats is the only possibility for Sphagnum colonization (Pfadenhauer and Klötzli, 1996). Nevertheless, fast growing vascular plants can support peat moss growth by diminishing wave movement and offering adherence area (Sliva, 1997). Besides the need for a preferably calm water surface, another limiting factor for floating mat growth is the water $CO_2$ concentration (Gaudig, 2002; Paffen and Roelofs, 1991; Smolders et al., 2001; Lamers, 2001; Lütt, 1992) which can be enhanced by vascular plants by providing oxygen to the rhizosphere fostering soil respiration. It thus seems conceivable that the Sphagnum spp. growth-favoring effects could outweigh the negative ramifications for bog development and climate change mitigation potential that the current plant cover implies. In sections of Himmelmoor with a non-industrial land use history, overgrowth of the grass tussocks, formerly dominating the area, by the bog-type Sphagnum species S. magellanicum and S. papillosum is in progress today (personal observation, 2016). The now prevailing plant species on the extraction site could therefore constitute an intermediate state that can potentially be overcome. The active dispersal of Sphagnum mosses as a management strategy would foster mire re-establishment and possibly lead to drastically diminished $CH_4$ release as e.g. the study from Järveoja et al. (2016) from an Estonian site where peat mosses dominate after rewetting suggests.

**Specific comments**

**Abstract**
Page 1, Line 18: The numbers in CO2 fluxes from rewetted and drained section are wrong. Rewetted section should have lower CO2 emission as stated in the manuscript.

True, numbers were switched around, order was reversed.

Page 1, Line 20: It is not useful to compare the difference in CH4 to the difference in CO2 in an absolute term. Surely CO2 is larger in the magnitude.

We suspect a misunderstanding. In this sentence not absolute $CH_4$ and $CO_2$ fluxes are compared but absolute differences in $CH_4$ fluxes from both land use types.

**Introduction**
Page 2, Line 27-29: Please provide references.

Still refers to Couwenberg et al. 2010 (see page 2, line24).

Page 3, Line 1: What do you mean by "higher plants"?

We mean "vascular plants" and agree that "higher plants" is ambiguous and therefore replaced this expression.

**Material and methods**

Page 5, Line 19: So the drained section had also some area rewetted during the study period? Please specify what you meant here.

Ditch blocking was performed on small sections of the drained area after a final harvest in this year (June/July) and therefore after the investigation period of this study. We added a sentence clarifying this fact.

Peat harvesting on the eastern half continued until June 2018, rewetting of smaller sections in this area began, however, already in 2014 (after the investigation period of this study).

Page 7, Line 15: Salix spp..

Extra dot removed

Page 11, Line 1: Maybe replace "and" with "or"

No occurrence of "and" in the mentioned line, no change made.

Page 11,Line 13:"...70 % at all flux gaps that resulted from data division". What does that mean?

At times when "tower view" fluxes were mostly associated to for example the drained section, the surface class specific time series of the rewetted section have a gap (data division). To later on gap-fill the flux time series, a contribution of the rewetted section of 70 % was prescribed for these gaps.

Page 11, Line 15-17: Why not using median? Maybe a probability density function plot would justify your method for gapfilling CCveg.

We chose the value of the most frequent class because it is closer to the actual fraction of vegetated areas within the investigation area. The new Figure XX3 illustrates this fact.

**Results and discussion**

Page 14, Line 12-20: Was the ANN model prediction compared with the testing subsets as it should be? As written in Appendix A, "70 % training and 30 % validation data", it seems there is no testing subset of data.

Yes, that is true. We did not use a testing subset during optimization of the individual networks (1000 per ensemble). Due to rigorous quality filtering and data division (into rewetted and drained), data availability, especially in case of $CH_4$ fluxes, was limited and we decided to increase the number of training samples to improve training quality at the expense of reserving data points for testing in each individual run. We, however, conducted a similar type of validation (see Figure D3) by driving models that were optimized using Year 1 measurement data as targets with Year 2 environmental data and comparing the results to measured Year 2 gas fluxes. The allocation of testing data is commonly is implemented in ANN optimization in order to generate a data set to later on estimate model quality on data independent from optimization. We believe that we achieved a test of similar meaningfulness by exploiting the fact that we had two independent years of data to work with and could at the same time improve model optimization by increasing the sizes of the training and validation data sets.

Page 20, Line 20: There are many more recent studies on that topic. e.g. "Impact of water table level on annual carbon and greenhouse gas balances of a restored peat extraction area", Jarveoja et al., 2016, Biogeosciences.

We extended Section 3.3 with a discussion about the recommended publication and others.

..., which is supplied to these areas from the underlying aquifer. Figure XX2 illustrates the dependence of $F_{CH4}$ on the relative contribution of the vegetated strips to the EC footprint. Mean summer fluxes were significantly (Two-sample Kolmogorov-Smirnov test, $p < 0.01$) higher from the vegetated (67 nmol m$^{-2}$ s$^{-1}$) than from the bare (29 nmol m$^{-2}$ s$^{-1}$) areas. These results are in line with estimates from Vybornova (2017) who determined a mean annual $F_{CH4}$ of 50 nmol m$^{-2}$ s$^{-1}$ for the same vegetated strips in Himmelmoor with manual chambers. Vybornova et al. (2019) report mean annual $F_{CH4}$ from the bare peat areas of 10

nmol m$^{-2}$ s$^{-1}$. Further evidence for the decisive role the type of vegetation which is established after rewetting has on the magnitude of CH$_4$ release is provided by Järveoja et al. (2016). The authors report annual CH$_4$ budgets of 0.25 and 0.16 g m$^{-2}$ a$^{-1}$ at subsections of their site with relatively high and low water table respectively. The site Järveoja et al. (2016) investigated is the former peat extraction area Tässi in central Estonia (58° N). In contrast to Himmelmoor, restoration measures at this site included the active establishment (dispersal) of peat mosses on a substantial layer (2.5 m) of remnant Sphagnum spp. peat. By the time measurements commenced two years after first restoration efforts were made, Tässi was already dominated by Sphagnum spp. mosses. With a lack of aerenchymatic plants and systematic efforts to re-establish bog vegetation, annual CH$_4$ release at Tässi is up to 100 times smaller than at Himmelmoor.

**Conclusions**

Page 21, Line 16-17: "The release of CH4 increases after rewetting and within the present two year data set also over time." This sentence does not read very well.

We agree, section reformulated, see comment below.

Page 21, Line 17-18: This statement does not correspond to the current results. CO$_2$ decreased from 887 to 567 g m-2 yr-1 while CO2e of CH$_4$ increased from 453 to 621 g m-2yr-1 in the rewetted section from year 1 to year 2. Otherwise, comparing the rewetted to the drained section, CO2 dropped from 974 to 567 g m-2 yr-1 and CO$_2$e of CH$_4$ increased from 412 to 621 g m-2 yr-1 in year 2. Either way it showed the reduction of CO$_2$ emission was more prominent than the increase of CH$_4$ emission during rewetting.

We agree, the sentence is not written very clearly; "on short timescales" is ambiguous. Section replaced with:

CO$_2$ emissions decreased progressively after rewetting with a reduction of 101 g m$^{-2}$a$^{-1}$ in Year 1 and of 407 g m$^{-2}$a$^{-1}$ in Year 2. The release of CH$_4$-CO2$_e$ increased after rewetting and was constant in both investigated years (209 g m$^{-2}$a$^{-1}$). The climate impact of elevated CH$_4$ emissions after rewetting therefore dominated over the effect of decreasing CO$_2$ release in Year 1, whereas CO$_2$ emission reduction was nearly twice as high as the CH$_4$-CO$_2$e increase in Year 2.

**Technical comments**

1) Maybe some of the figures D1-4 can be moved to the main text as they validated the modelling and the flux decomposition method.

We propose to move figures D3 and D4 to the results section as they show the most independent and therefore meaningful test of model quality.

2) The results on the cumulative fluxes were repeatedly presented in multiple units (g m-2a-1 in Table 2, mol m-2, CO2-C g m-2 and CH4-C g m-2 in Figure 5). Maybe Table 2 and figure 5 can be combined instead.

The purpose of reporting different units was to facilitate quick comparability to other studies. We agree that spreading the information across Table 2 and Figure 5 is not ideal. We therefore added carbon fluxes to Table 2. Figure 5 was left because there is a reference to the shape of the cumulative curve in the text (Page 18, Line 29).

**Table 2.** Annual sums of half-hourly carbon dioxide ($CO_2$) and methane ($CH_4$) fluxes from the drained and rewetted sections of the peat extraction site in Himmelmoor. $CH_4$ fluxes are expressed as $CO_2$ equivalents ($CO_2e$) using a global warming potential of 34 referring to a 100-year time horizon following Myhre et al. (2013). Year 1: 01 June 2012 to 31 May 2013; Year 2: 01 June 2013 to 31 May 2014

| | | Cumulative flux, g m$^{-2}$ a$^{-1}$ | |
| | | Surface class *drained* | Surface class *rewetted* |
|---|---|---|---|
| $CO_2$ | Year 1 | 988 ± 247 | 887 ± 296 |
| | Year 2 | 974 ± 292 | 567 ± 263 |
| $CH_4$ | Year 1 | 7.2 ± 1.8 | 13.3 ± 1.9 |
| | Year 2 | 12.1 ± 1.4 | 18.3 ± 1.5 |
| $CH_4$-$CO_2e$ | Year 1 | 244 ± 61 | 453 ± 63 |
| | Year 2 | 412 ± 46 | 621 ± 51 |
| total $CO_2e$ | Year 1 | 1232 ± 308 | 1340 ± 359 |
| | Year 2 | 1386 ± 338 | 1188 ± 314 |
| $CO_2$-C | Year 1 | 269 ± 67 | 242 ± 81 |
| | Year 2 | 266 ± 80 | 155 ± 72 |
| $CH_4$-C | Year 1 | 5.4 ± 1.4 | 10.0 ± 1.4 |
| | Year 2 | 9.1 ± 1.1 | 13.7 ± 1.1 |

**References**

Matthes, Jaclyn Hataa and Sturtevant, Cove and Verfaillie, Joseph and Knox, Sara and Baldocchi, Dennis: Parsing the variability in CH4 flux at a spatially heterogeneous wetland: Integrating multiple eddy covariance towers with high-resolution flux footprint analysis, JOURNAL OF GEOPHYSICAL RESEARCH-BIOGEOSCIENCES, 119, 7, 1322-1339, 2014.

[revised manuscript text omitted]